# SpikeStereoNet: A Brain-Inspired Framework for Stereo Depth Estimation from Spike Streams

**Zhuoheng Gao**[1], **Yihao Li**[1,4], **Jiyao Zhang**[2], **Rui Zhao**[1], **Tong Wu**[1], **Hao Tang**[1],
**Zhaofei Yu**[1,3], **Hao Dong**[2*], **Guozhang Chen**[1*], **Tiejun Huang**[1]

[1] National Key Laboratory for Multimedia Information Processing,
 School of Computer Science, Peking University
[2] Center on Frontiers of Computing Studies, School of Computer Science, Peking University
[3] Institute for Artificial Intelligence, Peking University
[4] School of Engineering and Applied Science, University of Pennsylvania
https://github.com/Criticality-Cognitive-Computation-Lab/SpikeStereoNet

## ABSTRACT

Conventional frame-based cameras often struggle with stereo depth estimation in rapidly changing scenes. In contrast, bio-inspired spike cameras emit asynchronous events at microsecond-level resolution, providing an alternative sensing modality. However, existing methods lack specialized stereo algorithms and benchmarks tailored to the spike data. To address this gap, we propose SpikeStereoNet, a brain-inspired framework to estimate stereo depth directly from raw spike streams. The model fuses raw spike streams from two viewpoints and iteratively refines depth estimation through a recurrent spiking neural network (RSNN) update module. To benchmark our approach, we introduce a large-scale synthetic spike stream dataset and a real-world stereo spike dataset with dense depth annotations. SpikeStereoNet outperforms existing methods on both datasets by leveraging spike streams' ability to capture subtle edges and intensity shifts in challenging regions such as textureless surfaces and extreme lighting conditions. Furthermore, our framework exhibits strong data efficiency, maintaining high accuracy even with substantially reduced training data.

## 1 INTRODUCTION

Depth perception is fundamental for navigating and interacting with the 3D world (Tosi et al., 2025), driving applications from robotic manipulation (Ma et al., 2024; Li et al., 2024; Wen et al., 2025b; Gao et al., 2024) to autonomous navigation (Wei et al., 2024; Duba et al., 2024; Kalenberg et al., 2024; Nahavandi et al., 2025; Wu et al., 2023; Xu et al., 2023c; Wang et al., 2024b). Traditional stereo vision estimates depth from calibrated image pairs captured by frame-based cameras; however, it suffers from motion blur and latency in dynamic scenes. Biological vision systems efficiently process visual inputs using sparse, asynchronous spikes, achieving remarkable speed and energy efficiency (Kundu et al., 2021; Yang et al., 2024; Datta et al., 2021; Göltz et al., 2021; Wang et al., 2025; Wolf & Lappe, 2021; Kucik & Meoni, 2021). Neuromorphic spike cameras (Zhao et al., 2024c; Zhang et al., 2022a; Zhao et al., 2021a;b; Hu et al., 2022; Zhao et al., 2022; Chen et al., 2024a; Zhao et al., 2024a) implement this principle in hardware, delivering ultra-high temporal resolution (up to 40,000 Hz) and capturing rich luminance information through asynchronous binary spike streams. This makes them suitable for robust perception tasks, such as stereo depth estimation, particularly in highly dynamic scenarios where conventional methods are ineffective.

Despite their potential, spike cameras present distinct challenges for stereo depth estimation. Their asynchronous, binary, high-throughput streams conflict with frame-based algorithms that expect synchronous, intensity-valued image pairs (Tankovich et al., 2021; Lipson et al., 2021; Li et al., 2022; Zhao et al., 2023; Rao et al., 2023; Xu et al., 2023a; Zeng et al., 2023; Guan et al., 2024;

---

*Corresponding authors.

Wang et al., 2024a; Xu et al., 2023b; Weinzaepfel et al., 2023; Chen et al., 2024b; Cheng et al., 2025; Jiang et al., 2025; Wen et al., 2025a), and any conversions to frames introduce temporal quantization errors, motion blur, or significant computational overhead (Zhao et al., 2021b; 2022; Chen et al., 2023), diminishing the sensor's intrinsic advantages. Although event-based stereo methods have been developed for DVS cameras (e.g. (Zhou et al., 2021; Zhang et al., 2022b; Cho et al., 2021; Lou et al., 2024; Rançon et al., 2022)), they rely on temporal contrast, while spike cameras emit integrated intensity streams that require tailored processing techniques. Critically, the field lacks specialized algorithms and benchmarks designed for stereo depth estimation directly from raw spike streams, hindering progress in this promising field.

Recent advances in spiking neural networks (SNNs) provide the foundation for our models design. Classical neuron models offer compact yet expressive dynamics for spike-based computation (Izhikevich, 2003). Building on these, surrogate-gradient methods enable scalable training of deep SNNs with gradient-based optimization (Neftci et al., 2019). Adaptive recurrent SNNs further improve temporal modeling and efficiency in time-domain tasks (Yin et al., 2021). In parallel, neuromorphic vision research demonstrates the benefits of spike-driven processing for object perception and recognition, highlighting the potential of spike-based architectures in real-world vision applications (Bi et al., 2019). Beyond SNNs-based, some work explores biologically inspired network architectures for vision. ClearSight (Lin et al., 2025) proposes a dual-drive hybrid model with neuron and synapse-based attention for event-based motion deblurring. SABV-Depth (Wang et al., 2023) integrates bio-inspired attention into a monocular depth network to enhance prediction accuracy. Earlier cortical models for object recognition, such as the neocognitron and hierarchical feedforward architectures (Fukushima, 1980; Serre et al., 2007), further demonstrate how principles from visual neuroscience can guide the design of robust perception systems.

To bridge this gap, we introduce SpikeStereoNet (Fig. 1), an end-to-end, biologically inspired framework for generating stereo depth directly from raw spike streams. SpikeStereoNet integrates a recurrent spiking neural network (RSNN) into an iterative refinement loop (Teed & Deng, 2020) and models neuronal interactions (Henry, 1995) to capture spatiotemporal dynamics inherent in spike data. The analysis of neuronal dynamics confirms the temporal stability, convergence, and expressive feature separation of the RSNN components. We introduce two novel benchmark datasets: a large-scale synthetic dataset with diverse scenes and ground-truth depth, and a real-world dataset containing synchronized stereo spike streams and corresponding depth sensor measurements. Experimental results demonstrate that SpikeStereoNet generalizes effectively, maintaining robust performance even when trained with substantially limited data. Our main contributions are as follows:

- We present the large-scale synthetic and real-world raw spike stream datasets for stereo depth estimation to offer an evaluation benchmark for this emerging field.

- We propose a novel biologically inspired RSNN-based SpikeStereoNet architecture that refines asynchronous spike data through iterative updates.

- We analyze the dynamics of RSNN iterations to demonstrate the stability and convergence properties of the model.

- We demonstrate the data efficiency of the proposed framework, highlighting the strong generalization even with limited training samples.

## 2 RELATED WORK

**Spike Cameras and their Applications.** Neuromorphic cameras, including event cameras (Moeys et al., 2018; Posch et al., 2011; Huang et al., 2017) and spike cameras (Huang et al., 2023), are inspired by the primate retina and operate asynchronously at the pixel level, providing ultra-high temporal resolution, wide dynamic range, low latency, and reduced energy consumption. Unlike event cameras, which follow a differential sampling model and capture only luminance changes in logarithmic space, spike cameras employ an integral sampling model, accumulating photons until a threshold is reached and then firing spikes. Although event cameras tend to generate sparser output, they might lose scene textures, especially in static regions due to their change-based design. In contrast, spike cameras can capture richer spatial details, making them well suited for tasks such as high-speed imaging (Zhao et al., 2024a), 3D reconstruction (Zhang et al., 2024b), motion deblurring (Zhang et al., 2024d), optical flow (Zhao et al., 2022; 2024b; 2025; Xia et al., 2023), object detection

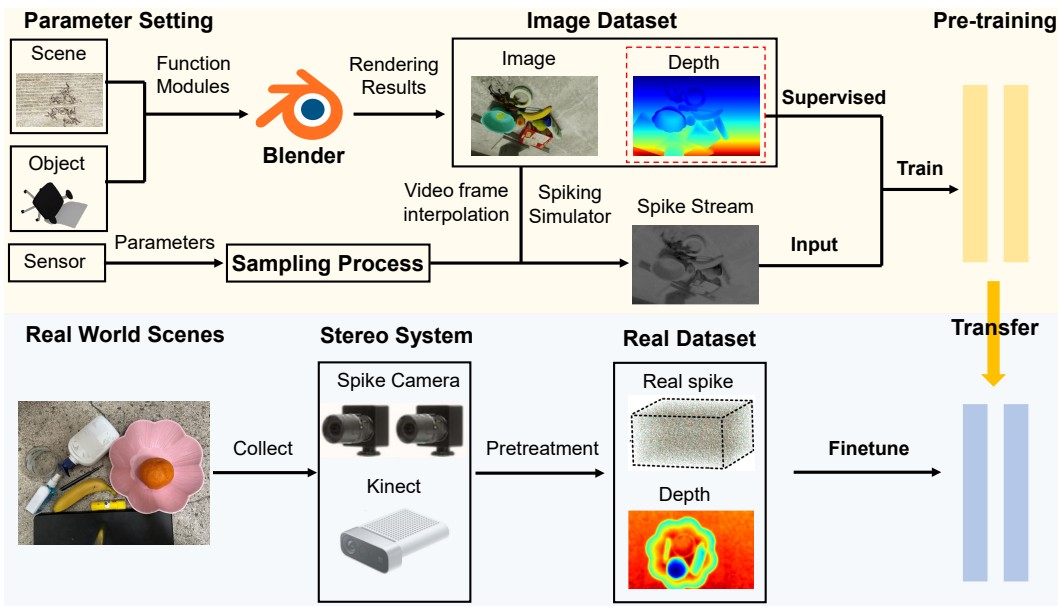

Figure 1: The overall pipeline is illustrated in the figure below. The upper path represents the training and evaluation process on synthetic dataset, while the lower path shows the transfer learning and testing procedure on real spike stream data by using the pre-trained model.

(Zheng et al., 2023), occlusion removal (Zhang et al., 2024c), monocular depth estimation (Zhang et al., 2022a), and binocular depth estimation (Wang et al., 2022).

**Frame-Based Stereo Depth Estimation.** Deep learning has significantly advanced stereo matching, leading to various methods with enhanced accuracy and generalization. The early methods (Fan et al., 2021; Yin et al., 2019) constructed cost volumes for each disparity and applied 3D CNNs for refinement. In contrast, recurrent refinement-based models (Rao et al., 2023; Tian et al., 2023) inspired by RAFT-Stereo (Lipson et al., 2021; Teed & Deng, 2020) iteratively update the disparity without building a full 4D volume. It restricts 2D flow to the 1D disparity dimension and uses multi-level convolutional Gated Recurrent Unit (ConvGRU) for broader receptive fields, demonstrating strong generalization. Successor models such as DLNR (Zhao et al., 2023), IGEV-Stereo (Xu et al., 2023a) and Selective-Stereo (Wang et al., 2024a) improve in-domain performance, but RAFT-Stereo retains advantages in zero-shot settings (Lipson et al., 2021). Recent hybrid methods (Hamid et al., 2021) combine cost filtering and iterative refinement for better trade-offs. Meanwhile, transformer-based solutions (Xu et al., 2023b) utilize cross-attention or global self-attention.

**Event-Based Stereo Depth Estimation.** Event cameras capture pixel-level brightness changes asynchronously, offering significant advantages over conventional frame-based cameras. Recently, event-based stereo depth estimation has advanced rapidly (Gallego et al., 2022; Tosi et al., 2025), with notable methods exploiting camera velocity (Zhang et al., 2022b) or deriving depth without explicit event matching (Zhou et al., 2018). Deep learning approaches propose novel sequence embedding (Tulyakov et al., 2019) and fuse frame and event data (Nam et al., 2022) to improve depth estimates in challenging scenarios. Moreover, some works (Cho et al., 2023) leverage standard models from the image domain, exploiting the inherent connection between frame and event data to improve stereo depth estimation performance.

## 3 METHODS

This section describes the overall architecture of our proposed model (Fig. 2). Figs. 2a, b illustrate the design of the biologically inspired framework. Fig. 2c shows the overall network architecture, which consists of the following four main components.

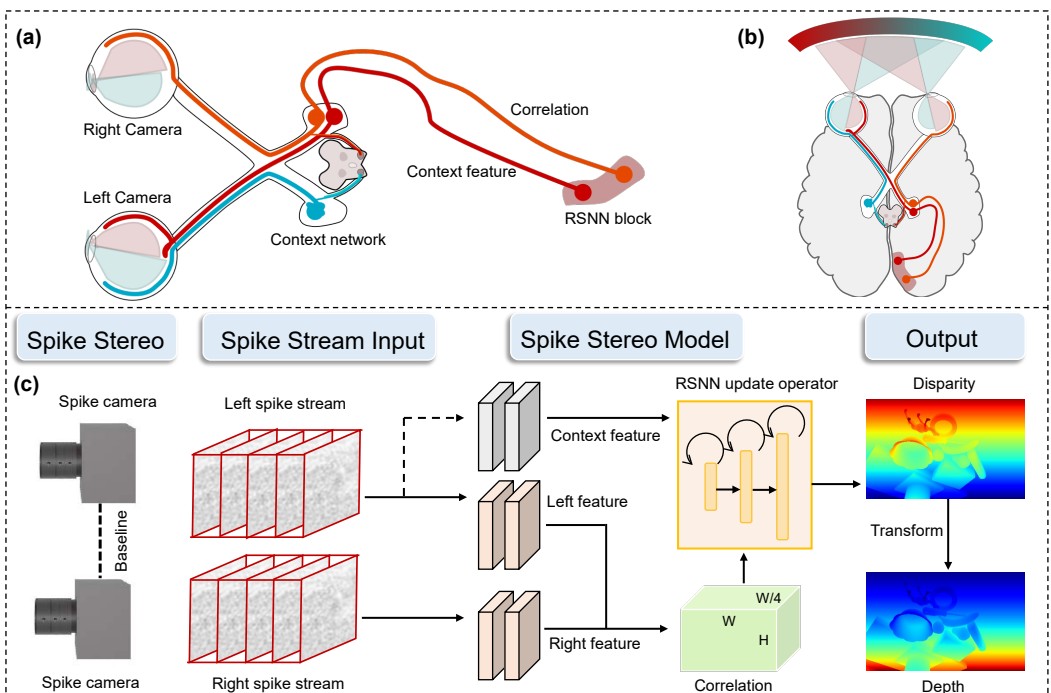

Figure 2: The pipeline of the proposed solution. **(a)** and **(b)** The components of the biological visual system, each corresponding to a specific module in the computational framework. **(c)** The overview of SpikeStereoNet: Multi-scale spike features are first extracted to construct a correlation pyramid, followed by a biologically inspired RSNN-based update operator that iteratively refines disparity using local cost volumes and contextual cues. The final disparity map is upsampled to produce high-resolution depth estimates.

**Spike Camera Model.** The spike camera, mimicking the retinal fovea, consists of a $H \times W$ pixel array in which each pixel asynchronously fires spikes to report luminance intensity. Specifically, each pixel independently integrates the incoming light over time. At a given moment $t$, if the accumulated brightness at pixel $(i, j)$ reaches or exceeds a fixed threshold $C$, a spike is fired and the accumulator is reset: $A(i, j, t) = \int_{t_s}^{t} I(i, j, \tau) \, d\tau \geq C$, where $i, j \in \mathbb{Z}, i \leq H, j \leq W$. Here, $A(i, j, t)$ denotes the accumulated brightness at time $t$, $I(i, j, \tau)$ is the instantaneous brightness at the pixel $(i, j)$, and $t_s$ is the last firing time of the spike. If $t$ corresponds to the first spike, then $t_s = 0$. Due to circuit-level constraints, spike readout times are quantized. Although firing is asynchronous, all pixels periodically check their spike flags at discrete times $n\Delta t$ ($n \in \mathbb{Z}$), where $\Delta t$ is a short interval on the order of microseconds. As a result, each readout forms a binary spiking frame of size $H \times W$. Over time, these frames constitute a $H \times W \times N$ binary spike stream:

$$S(i, j, n\Delta t) = \begin{cases} 1 & \text{if } \exists t \in [(n-1)\Delta t, n\Delta t] \text{ s.t. } A(i, j, t) \geq C, \\ 0 & \text{if } \forall t \in [(n-1)\Delta t, n\Delta t] \text{ s.t. } A(i, j, t) < C, \end{cases} \tag{1}$$

then we denote the generated spike stream as $S \in \{0, 1\}^{N \times H \times W}$, where $H$ and $W$ signify the height and width of the image, and $N$ represents the temporal steps of the spike stream.

**Spike Feature Extraction.** The structure of spike feature extraction is illustrated in Fig. 3(a), which outlines the architecture of the feature network or the context network. The feature network receives the left and right input spike streams $S_{l(r)} \in \mathbb{R}^{N \times H \times W}$, applying a $7 \times 7$ convolution to downsample them to half-resolution. Then, a series of residual blocks is used for spike feature extraction, followed by another downsampling layer to obtain 1/4-resolution features. For more expressive representations, we further generate multi-scale features at 1/4, 1/8, and 1/16 scales, denoted $\{f_{l,i}, f_{r,i} \in \mathbb{R}^{C_i \times H_i \times W_i}\}, i \in \{4, 8, 16\}$. Among these, $f_{l,4}$ and $f_{r,4}$ are used to construct the cost volume, while all levels are used as a guide for the 3D regularization network.

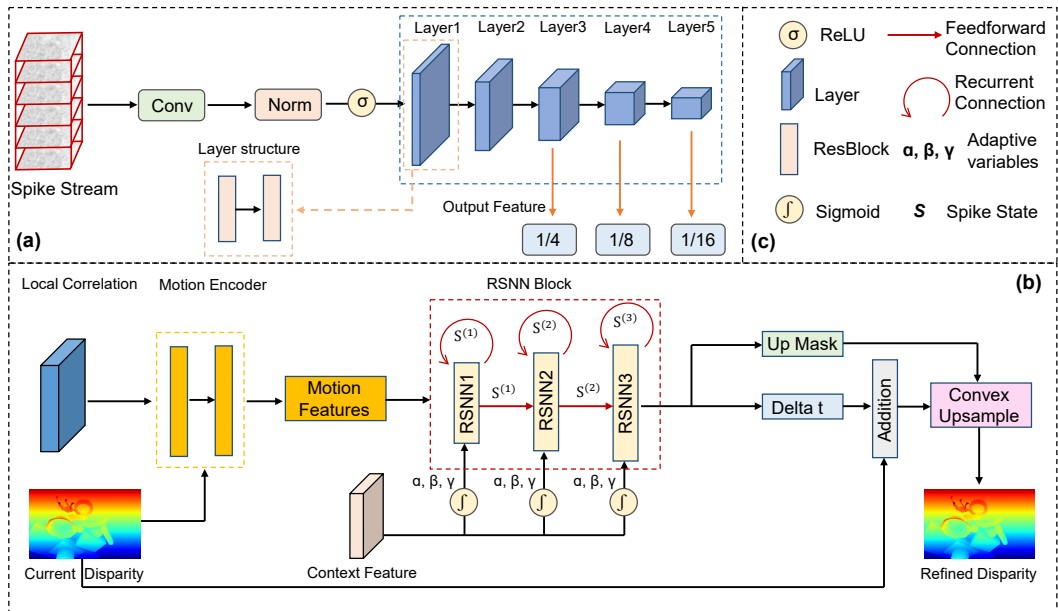

Figure 3: Illustration of the detailed structure of spike feature extraction and the RSNN-based update module. **(a)** Spike feature extraction: It comprises one context network and two feature networks, which extract multi-scale correlation features, contextual features, and the initial hidden state from the spike streams. A single network structure is illustrated in the diagram. **(b)** RSNN-based update block: Local correlations and disparity fields are used to generate motion features, which update the RSNN hidden states through recurrent and feedforward connections. The RSNN at the highest resolution is responsible for refining the disparity estimates. **(c)** Descriptions of key modules.

The context network shares a similar backbone with feature networks. It consists of residual blocks and additional downsampling layers to produce multi-scale context features of the original resolution. These features are used to initialize and inject contexts into RSNNs at each recurrent iteration.

**Correlation Volume.** Given the left and right feature maps $f_{(l)}$ and $f_{(r)}$, we first construct a correlation cost volume of all pairs: $C_{ijk} = \sum_h f_{(l)hij} \cdot f_{(r)hik}$. Next, we build a 4-level correlation pyramid $\{C_i\}_{i=1}^4$ applying 1D average pooling along the last dimension. Each level is obtained using a kernel size of 2 and a stride of 2, progressively downsampling the disparity dimension.

**RSNN-Based Update Operator.** To capture multi-scale spatial and temporal information, a three-layer RSNN is used as the update module. Each RSNN layer contains recurrent intra-layer connections and feedforward inter-layer connections to facilitate hierarchical temporal processing (Fig. 3b, c). At 1/8 and 1/16 resolutions, RSNN modules receive inputs from the context features, the post-synaptic currents obtained by weights from the spike states at the same layer and the previous layer. These recurrent and feedforward connections expand the receptive field and facilitate feature propagation across scales. At 1/4 resolution level, the model additionally receives the current disparity estimate and the local cost volume derived from the correlation pyramid. A single RSNN updates as follows (the default parameters are the $l$-th layer network, the parameters from the previous layer are explicitly indicated, and bold symbols represent vectors/matrices):

$$\begin{aligned}
\boldsymbol{\alpha_t} &= \sigma(\mathrm{Conv}([\boldsymbol{s_{t-1}}, \boldsymbol{x_t}], W_\alpha) + \boldsymbol{c_\alpha}), \\
\boldsymbol{\beta_t} &= \sigma(\mathrm{Conv}([\boldsymbol{s_{t-1}}, \boldsymbol{x_t}], W_\beta) + \boldsymbol{c_\beta}), \\
\boldsymbol{\gamma_t} &= \sigma(\mathrm{Conv}([\boldsymbol{s_{t-1}}, \boldsymbol{x_t}], W_\gamma) + \boldsymbol{c_\gamma}), \\
\boldsymbol{s_t} &= f(\boldsymbol{s_{t-1}}, \boldsymbol{s_t^{(l-1)}}, \boldsymbol{\alpha_t}, \boldsymbol{\beta_t}, \boldsymbol{\gamma_t}),
\end{aligned} \qquad (2)$$

where $\boldsymbol{s_t}$ is the spike state in layer $l$ and timestep $t$, $\boldsymbol{c_\alpha}$, $\boldsymbol{c_\beta}$, and $\boldsymbol{c_\gamma}$ are context embeddings from the context network, and $\sigma$ is the sigmoid function. The number of channels in the hidden states of RSNNs is 128, same as in the context feature. The function $f(\cdot)$ corresponds to our proposed

adaptive leaky integrate-and-fire neuron model, defined as:

$$
\begin{aligned}
\boldsymbol{h_t} &= \boldsymbol{\alpha_t} \cdot \boldsymbol{v_{t-1}} + (1 - \boldsymbol{\alpha_t}) \cdot (W_{\text{rec}} \boldsymbol{s_{t-1}} + W_f \boldsymbol{s_t^{(l-1)}}), \\
\boldsymbol{v_t^{\text{th}}} &= \boldsymbol{\beta_t} \cdot v_{\text{peak}}, \\
\boldsymbol{s_t} &= \theta(\boldsymbol{h_t} - \boldsymbol{v_t^{\text{th}}}), \\
\boldsymbol{v_t} &= \boldsymbol{h_t} - \boldsymbol{\gamma_t} \cdot \boldsymbol{s_t} \cdot \boldsymbol{v_t^{\text{th}}},
\end{aligned}
\tag{3}
$$

where $\boldsymbol{v_t}$ is the membrane potential, $\boldsymbol{v_t^{\text{th}}}$ is the firing threshold and $\theta(\cdot)$ is the Heaviside step function. $W_{\text{rec}}$ and $W_f$ are the recurrent and feedforward synaptic weights, respectively, implemented as convolutional kernels. And $\boldsymbol{\alpha}$, $\boldsymbol{\beta}$ and $\boldsymbol{\gamma} \in [0, 1]$ are adaptive variables, representing retention of membrane potential, firing threshold, and soft reset in the previous step, respectively. This update mechanism draws biological inspiration from neuroscience findings showing that key neuronal properties such as firing threshold, resting potential, and membrane time constant are not fixed but vary dynamically in biological neural circuits depending on neuronal state and contextual input.

**Spatial Upsampling.** Based on the hidden state of final layer, network predicts both residual disparity through two convolutional layers and then updates current disparity: $d_t = d_{t-1} + \Delta d_t$. Finally, the disparity at 1/4 resolution is upsampled to full resolution using a convex combination strategy.

**Loss Function.** We supervise the network for a composite loss function composed of the three terms: the main loss, the firing rate and the voltage regularization. The overall loss is defined as:

$$
\begin{aligned}
L &= L_{\text{stereo}} + \lambda_f L_{\text{rate\_reg}} + \lambda_v L_{\text{v\_reg}} \\
&= \sum_{t=1}^{T} \eta^{T-t} \|d_{\text{gt}} - d_t\|_1 + \lambda_f \sum_{i=1}^{N} (r_i - r_0)^2 + \lambda_v \sum_{i=1}^{N} \sum_{t=1}^{T} v_i(t)^2, \quad \eta = 0.9.
\end{aligned}
\tag{4}
$$

The first term $L_{\text{stereo}}$ is the main loss, which is the $L_1$-norm distance between all predicted disparities $\{d_t\}_{t=1}^{T}$ and the ground truth disparity $d_{\text{gt}}$ with increasing weights. To constrain the firing rate of neurons and promote temporal sparsity, the second term $L_{\text{rate\_reg}}$ is firing rate regularization, $r_i$ is the average firing rate of the $i$-th neuron during time $T$, and $r_0$ is the target firing rate. Furthermore, the third term $L_{\text{v\_reg}}$ serves as the voltage regularization term, where $N$ represents the total number of neurons. Both regularization terms encourage sparsity and benefit the performance.

# 4 EXPERIMENTS

## 4.1 DATASETS

**Synthetic Dataset.** To support supervised training for spike-based depth estimation, we construct a high-quality synthetic dataset using Blender. The dataset includes 150 training scenes and 40 testing scenes, each containing 256 pairs of RGB images and corresponding ground-truth depth maps. The dataset build process and details are in the Appendix A.3.

To generate dense spike streams with high temporal resolution in dynamic environments, we apply a video interpolation method (Zhang et al., 2023) to synthesize 50 intermediate RGB frames between each pair of adjacent frames. These high-frequency image sequences are then converted into spike streams using a biologically inspired spike generation mechanism that models the analog behavior of real spiking camera circuits. Noise modeling is also integrated to further enhance realism. Our dataset provides synchronized high-resolution spike streams and pixel-wise depth ground truth, offering a comprehensive and reliable benchmark for training and testing.

**Real Dataset.** We collect a real-world spike stereo dataset using two spike cameras and one depth camera (Kinect) in diverse environments to evaluate the generalization and practical performance of the model. The spike cameras have a resolution of $400 \times 250$ with a temporal resolution of 20,000 Hz, while the depth camera provides RGB-D data at $1280 \times 720$ resolution and 30 FPS. The captured scenes involve various objects of the real world recorded in indoor settings. The dataset includes over 3,000 stereo spike stream pairs (more than 300,000 frames) and depth maps and is divided into training and test sets. More details about our real dataset and system are provided in Appendix A.4.

Table 1: Quantitative results on the test set of synthetic dataset, and all methods use spike streams as input. Best results for each evaluation metric are bolded, second best are underlined. The "↓" indicates that the lower the metrics, the better.

| Method | bad 1.0 (%) ↓ | bad 2.0 (%) ↓ | bad 3.0 (%) ↓ | AvgErr (px) ↓ | Params (M) | FLOPs (B) |
|---|---|---|---|---|---|---|
| Stereospike (Rançon et al., 2022) | 14.10 | 9.82 | 5.35 | 1.10 | **5.10** | 569 |
| ZEST (Lou et al., 2024) | 11.10 | 4.94 | 3.50 | 0.62 | 340.75 | 989 |
| CREStereo (Li et al., 2022) | 16.12 | 10.54 | 4.40 | 0.81 | 5.43 | 863 |
| RAFT-Stereo (Lipson et al., 2021) | 9.67 | 4.64 | 2.76 | 0.48 | 11.41 | 798 |
| GMStereo (Xu et al., 2023b) | 17.61 | 11.05 | 4.39 | 0.83 | 7.40 | **160** |
| IGEV-Stereo (Xu et al., 2023a) | 12.50 | 6.50 | 4.27 | 0.76 | 12.77 | 614 |
| DLNR (Zhao et al., 2023) | 10.15 | 4.67 | 2.81 | 0.55 | 57.83 | 1580 |
| MoCha (Chen et al., 2024b) | 11.93 | 5.67 | 4.39 | 0.73 | 20.96 | 935 |
| Selective (Wang et al., 2024a) | 9.24 | 4.57 | 2.66 | 0.45 | 13.32 | 957 |
| MonSter (Cheng et al., 2025) | 9.23 | 4.64 | 2.72 | 0.46 | 388.69 | 1567 |
| SpikeStereoNet (Ours) | **8.41** | **4.13** | **2.38** | **0.42** | 12.15 | 473 |

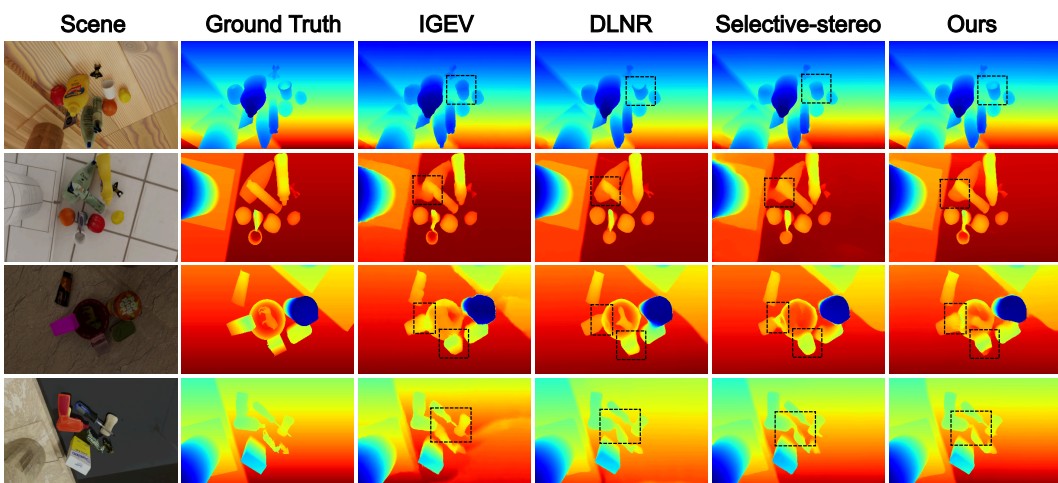

Figure 4: From left to right: synthetic scene images from the left view, ground-truth depth, depth prediction results from existing stereo methods and our method.

## 4.2 IMPLEMENTATION DETAILS

We implemented SpikeStereoNet using PyTorch and performed all experiments on NVIDIA RTX 4090 GPUs. The model is trained using the AdamW optimizer with gradient clipping in the range of $[-1, 1]$. A one-cycle learning rate schedule is adopted, with an initial learning rate set to $2 \times 10^{-4}$. During training, we apply 16 update iterations for each sample and set the batch size as 8 for 300k steps. We use random horizontal and vertical flips as data augmentation in the training dataset.

## 4.3 EXPERIMENTAL RESULTS

Our method is compared with various categories of stereo depth estimation approaches. First, event-based stereo networks are compared, such as ZEST (Lou et al., 2024). Next, state-of-the-art frame-based stereo networks are evaluated, including Selective-Stereo (Wang et al., 2024a), RAFT-Stereo (Lipson et al., 2021), and GMStereo (Xu et al., 2023b), etc. These methods cover different strategies, such as iterative refinement or transformer-based global matching. All networks are re-trained using the same settings and evaluated under identical conditions to ensure fair comparison.

**Experiments on the Synthetic Dataset.** We use the average end-point error (AvgErr) and the bad pixel ratio with different pixel thresholds to evaluate the quality of stereo depth estimation. The

Table 2: Quantitative results on the real dataset, and all methods use real spike streams as input.

| Method | bad 2.0 (%) ↓ | bad 3.0 (%) ↓ | AvgErr ↓ |
|---|---|---|---|
| RAFT-Stereo (Lipson et al., 2021) | 6.18 | 3.39 | 0.64 |
| IGEV-Stereo (Xu et al., 2023a) | 8.39 | 5.72 | 0.88 |
| Mocha-stereo (Chen et al., 2024b) | 7.32 | 5.88 | 0.81 |
| DLNR (Zhao et al., 2023) | 5.64 | 3.38 | 0.61 |
| Selective-Stereo (Wang et al., 2024a) | 5.50 | 3.43 | 0.58 |
| SpikeStereoNet (Ours) | **5.33** | **3.19** | **0.56** |

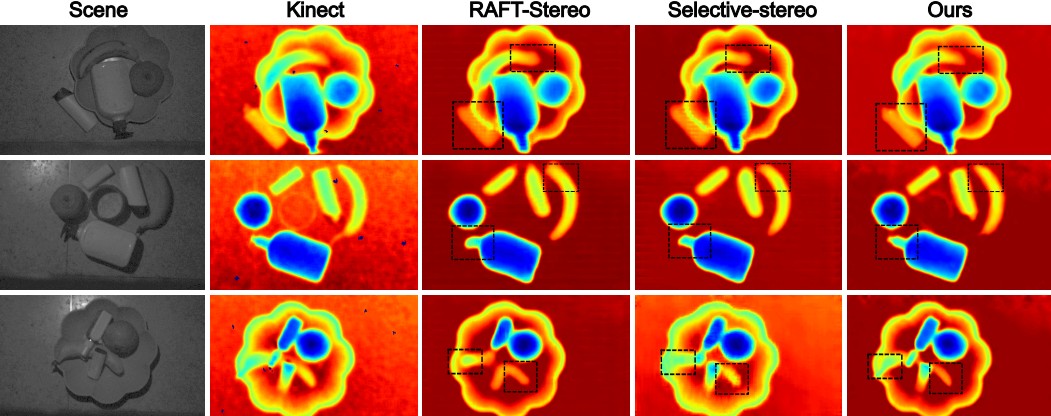

Figure 5: Visual results of our method and competing approaches on the real dataset. The "Scene" refers to the gamma-transformed temporal average of spike streams. "Kinect" represents the raw depths captured by the depth camera.

results of the quantitative comparisons are presented in Table 1. As shown in the table, we achieve the best performance among the listed methods for almost all metrics in the synthetic dataset. Fig. 4 illustrates the scene, the spike stream, the ground truth of the depth, and the predicted depth maps of different methods. Our method significantly outperforms competing approaches, producing more accurate and sharper depth results. The sharpness and density of predicted depth maps demonstrate the effectiveness of our approach in spike stream inputs, especially in complex and cluttered scenes.

In addition to accuracy metrics, Table 1 also reports the params and floating-point operations (FLOPs) for each method. SpikeStereoNet achieves a favorable balance between performance and efficiency, ranking second in FLOPs while outperforming most models in parameter number. This demonstrates that the additional computational cost, incurred to better capture the spatiotemporal characteristics and asynchrony of spike data is justified. Compared to other methods that handle complex temporal structures, such as Transformer-based approaches, SpikeStereoNet remains highly competitive in the efficiency. Additional comparison results are provided in Appendix A.5.

**Experiments on the Real Dataset.** To bridge the domain gap between synthetic and real data, we apply a domain adaptation strategy to fine-tune our model, which is initially trained on synthetic datasets. The visualization results are shown in Fig. 5. Compared with other methods, our approach produces more accurate depth maps with sharper object boundaries and finer structural details. It performs especially well in challenging regions, such as textureless surfaces and high-illumination areas, benefiting from the integration of spike domain characteristics. These results highlight the effectiveness of our domain adaptation strategy and the robustness of our model in real-world stereo depth estimation from spike streams. The detailed results of different models are shown in Table 2, and we can see that our method outperforms other methods with the smallest AvgErr in the real dataset. The results of experiments demonstrate that our method can effectively handle real scenes.

**Data Efficiency.** To evaluate the generalizability of our model, we conduct experiments by training it on randomly selected subsets of the synthetic dataset, using 10% to 50% of the full training data in

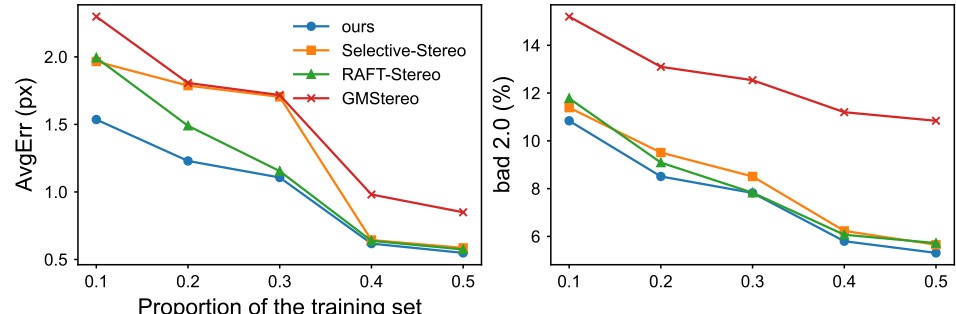

Figure 6: Data efficiency experiments on the synthetic dataset. The horizontal axis indicates the proportion of the training set, while the vertical axes represent the AvgErr and 2 px error, respectively.

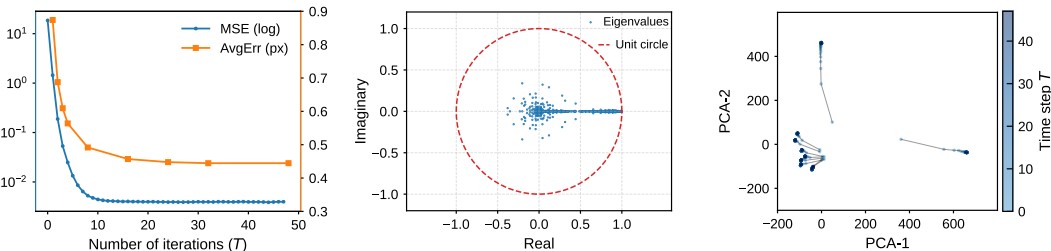

Figure 7: Visualization of network dynamics analysis. The sub-figures show, in order: the change in hidden state differences over time in the RSNN, the eigenvalue spectrum of the network weights for stability analysis, and the PCA of different inputs over time.

10% increments. All models, including ours and the baselines, are tested on the complete testing set to ensure a fair and consistent comparison. As illustrated in Fig. 6, our model consistently outperforms other methods in all training data ratios. The performance gap becomes more significant as the training data size decreases, demonstrating that our model generalizes better in data-scarce scenarios. These results highlight the robustness and efficiency of our approach in learning meaningful representations, even with limited supervision.

**Explanation of Iterative Refinement Structures.** We analyze the dynamic behavior of our hybrid neural network to assess its temporal characteristics. As shown in Fig. 7, the left panel illustrates that hidden state differences decrease over time, indicating convergence. The middle panel displays the eigenvalue spectrum of the Jacobian matrix of RSNN weights at the final time step, with all eigenvalues located within the unit circle, which confirms the stability of the system. The right panel presents a PCA of hidden states across input batches, showing increasing dispersion over time, which reflects the model's ability to encode diverse temporal patterns. These results highlight temporal stability, robustness, and expressive power of the network. Further theoretical proof can be found in the Appendix A.5.

Beyond validating our specific RSNN design, this dynamics analysis sheds light on why iterative refinement, the core mechanism in architectures like RAFT-Stereo (Lipson et al., 2021), is so effective. Our results explicitly show how a well-behaved iterative updater can maintain stability while progressively integrating information over time steps to refine the estimate towards the correct solution. This provides a dynamical grounding for the empirical success of iterative methods.

### 4.4 ABLATION STUDY

To evaluate the effectiveness of each component in our framework and analyze their contributions to the overall performance, we conducted a series of ablation studies in the synthetic dataset.

**Connection Structure.** We analyze the effect of connection structures within RSNN. When using feedforward connections (FFC) between layers or recurrent connections (RC) within layers, there

Table 3: Ablation studies of network architectures.

| Setting of experiment | bad 2.0 (%) ↓ | bad 3.0 (%) ↓ | AvgErr (px) ↓ |
|---|---|---|---|
| **w/o** RC & FFC | 12.07 | 6.46 | 1.29 |
| **w/o** RC | 11.05 | 4.48 | 0.83 |
| **w/o** FFC | 5.86 | 3.63 | 0.68 |
| **w/o** GN module | 7.49 | 3.11 | 0.58 |
| **w/o** voltage regularization | 7.33 | 2.97 | 0.57 |
| **w/o** firing-rate regularization | 6.38 | 2.85 | 0.51 |
| **w/o** regularization | 7.77 | 3.41 | 0.61 |
| **Ours (full)** | **4.13** | **2.38** | **0.42** |

Table 4: Ablation studies for the RSNN module.

| Method | bad 2.0 (%) ↓ | bad 3.0 (%) ↓ | AvgErr (px) ↓ |
|---|---|---|---|
| Vanilla RNN | 7.28 | 3.41 | 0.66 |
| GRU | 4.53 | 2.99 | 0.48 |
| LSTM | 4.77 | 2.94 | 0.49 |
| Raw SNN | 11.05 | 4.48 | 0.83 |
| LIF (fixed $\alpha$, $\beta$, $\gamma$) | 7.05 | 4.06 | 0.69 |
| **ALIF RSNN (Ours)** | **4.13** | **2.38** | **0.42** |

is a slight increase in runtime and parameter count. However, this structural enhancement leads to significantly better overall performance. As shown in the upper part of Table 3, incorporating these connections results in improved accuracy in all evaluation metrics. The improved information flow across time and layers enables networks to capture richer temporal dynamics and finer spatial details, highlighting the importance of these connections in achieving accurate stereo depth estimation.

**Regularization.** We investigate the impact of removing regularization (Reg) terms and group normalization (GN). As listed in the lower part of Table 3, removing either the regularization on firing rate and membrane potential or the group normalization applied to adaptive input variables leads to a clear drop in performance. The absence of regularization results in unstable training and increased prediction error, while removing group normalization slows convergence and reduces final accuracy. When both components are removed simultaneously, performance is further degraded.

**RSNN Structure.** We conducted explicit replacement experiments in which the adaptive LIF (ALIF) neurons used in our recurrent refinement were substituted with a classical LIF implementation that keeps all adaptive scalars fixed, where $\alpha$, $\beta$ and $\gamma$ are fixed constants, not learnable or input dependent. In addition, we also removed spikes and replaced them with vanilla RNN, LSTM, and GRU blocks, which have the same number of neurons as ALIF-RSNN. This removes adaptivity to temporal patterns in spike streams. We conducted an ablation study by replacing ALIF with the above models in the synthetic dataset. All other conditions of the experiment are the same, and the results are in Table 4. From the above table, it can be seen that after replacing the ALIF model, the overall performance significantly decreased. Adaptive gating markedly improves both stability and final accuracy with negligible overhead, justifying our choice of the novel ALIF.

## 5 CONCLUSION

We present SpikeStereoNet, the state-of-the-art deep learning framework for stereo depth estimation from spike streams of spike camera, with a biologically plausible update operator based on recurrent spiking neural networks (RSNNs). In addition, we construct synthetic and real-world stereo spike stream datasets with corresponding depth maps. Extensive experiments demonstrate that our method achieves SOTA performance, benefiting from the data efficiency and convergence of RSNN. Overall, SpikeStereoNet represents an advance in neuromorphic stereo vision and demonstrates the potential of spike-based systems for efficient, accurate, and high-speed 3D perception.

ETHICS STATEMENT

We affirm compliance with the Code of Ethics for all stages of this work. Our study does not involve interventions on human or animal subjects. We evaluate and report fairness indicators where applicable, and we avoid training or releasing models that systematically disadvantage protected groups. All code, models, and preprocessing scripts will be released with documentation to support reproducibility while respecting data licenses and usage constraints. We also report computing settings to encourage consideration of environmental impact and resource use.

REPRODUCIBILITY STATEMENT

To facilitate reproduction of our results, we provide detailed descriptions of our methodology. All experiments were conducted under PyTorch with CUDA, and the exact environment is specified. We specify all model architectures, hyperparameters, training protocols, data-splitting strategies, and evaluation metrics in the main body and the Appendix. All experiments were performed with fixed random seeds. We will release our complete dataset, code repository, and demo.

ACKNOWLEDGMENTS

This work was supported by the National Natural Science Foundation of China (Grant No. 62576011) and the PKU-AI$^2$ Robotics Joint Lab of Embodied AI.

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

# A  APPENDIX

The appendix provides additional implementation details and extended experimental results for the SpikeStereoNet framework introduced in the main paper. It is organized into six distinct sections: The Use of Large Language Models in Appendix A.1, Theory Analysis and Method in Appendix A.2, Details of the Synthetic Dataset in Appendix A.3, Details of the Real Dataset in Appendix A.4, More Experimental Results in Appendix A.5 and Broader Impacts in Appendix A.6.

## A.1  THE USE OF LARGE LANGUAGE MODELS (LLMS)

During writing, we use large language models only for grammar and style polishing. The LLMs did not generate or rewrite technical content. All statements and citations were verified by the authors. No proprietary data or restricted code was shared with the LLMs. The authors assume full responsibility for the content of the paper.

## A.2  THEORY ANALYSIS AND METHOD

### A.2.1  PROBLEM STATEMENT

Given a continuous binary spike stream $S_N^{t_1} \in \{0,1\}^{H \times W \times N}$ with a spatio-temporal resolution of $H \times W \times N$, centered on the timestamp $t_1$, we divide it into three non-overlapping substreams $S_N^{t_0}$, $S_N^{t_1}$, and $S_N^{t_2}$, centered on timestamps $t_0$, $t_1$, and $t_2$, respectively. The goal is to estimate the disparity map corresponding to $t_1$. Here, $S_N^{t_1}$ provides the essential spatial structure for estimating the disparity, while $S_N^{t_0}$ and $S_N^{t_2}$ provide temporal cues to improve consistency and robustness. Letting $S_n \in \{0,1\}^{W \times H}$ denote the $n$-th spike frame, we predict the disparity at timestamp $t_1$ from the full spike stream $\{S_n\}, n = 0, \ldots, k$, leveraging both spatial features and temporal continuity.

### A.2.2  NOISE SIMULATION

The noise in spiking cameras comes mainly from the dark current. The accumulation of brightness in the pixel $(i, j)$ over time $t$ can be expressed as:

$$A(i, j, t) = \int_{t_{i,j}^{\text{pre}}}^{t} \left( I(i, j, \tau) + I_{\text{dark}}(i, j, \tau) \right) d\tau, \tag{5}$$

where $A(i, j, t)$ represents the integrated brightness, $t_{i,j}^{\text{pre}}$ is the previous firing time for the pixel $(i, j)$, and $I_{\text{dark}}$ denotes the random noise component caused by the dark current. In our synthetic dataset, the dark currents in the circuits introduce thermal noise, which is the type of noise modeled in this work.

### A.2.3  VIDEO-TO-SPIKE PREPROCESSING PIPELINE

We introduce a two-stage preprocessing pipeline to convert conventional image data into temporally precise spike streams, comprising neural network-based frame interpolation and spike encoding.

**Frame Interpolation for Enhanced Temporal Resolution.** To enhance temporal resolution, raw frames from dynamic synthetic datasets are processed using a pre-trained video frame interpolation model EMA-VFI (Zhang et al., 2023). The architecture comprises a hybrid CNN and transformer framework, and uses correlation information hidden within the attention map to simultaneously enhance the appearance information and model motion.

For temporal expansion, we applied a frame rate upsampling factor of $\times 50$ to the synthetic dataset. The output is represented as a 4D tensor of shape $[T, H, W, C]$, where $T$ is the temporal length, $H \times W$ is the spatial resolution and $C = 3$ denotes RGB channels.

**Spike Encoding via Temporal Integration.** The high-frame-rate RGB sequences are converted to spike streams through a temporal integration algorithm:

1. Convert RGB frames to grayscale and normalize pixel intensities to $[0, 1]$.

2. Accumulate membrane potential over time as $V_t = V_{t-1} + I_t$.

3. Spike generation:

$$\text{spike matrix}[t, x, y] = \begin{cases} 1 & \text{if } V_t(x, y) \geq \theta \\ 0 & \text{otherwise} \end{cases}$$

where threshold $\theta = 5.0$ and potential reset $V_t \leftarrow V_t - \theta$ after spike.

4. Repeat until all frames are processed.

Spike streams are stored as binary tensors of shape $[T, H, W]$ using the `StackToSpike` function with configurable noise $I_{\text{noise}}$ and threshold $\theta$. For compact storage and compatibility for our framework, the `SpikeToRaw` module compresses spikes (8 per byte) in the `.dat` format, allowing lossless reconstruction during inference.

The proposed two-stage preprocessing pipeline effectively bridges conventional images and neuromorphic vision processing. By combining deep learning-based frame interpolation with bio-inspired spike encoding, we achieve the following:

- **Temporal Super-Resolution**: Neural interpolation extends the temporal sampling density by $50\times$ through multi-scale optical flow and attention mechanisms, preserving physical consistency in dynamic scenes.

- **Biologically Plausible Encoding**: The temporal integration algorithm emulates the neuron dynamics of the retina, converting intensity variations into sparse spike events with adaptive threshold control.

- **System Compatibility**: Serialized spike data (.dat) with byte-level compression and structured formatting ensure seamless integration into brain-inspired stereo depth estimation systems.

This pipeline facilitates the efficient transformation of dynamic scene datasets into spike-compatible formats, preserving configurable spatio-temporal characteristics and laying a solid foundation for spike-based stereo depth estimation.

To assess the fidelity of the conversion of the dataset, we use the Texture From Interval (TFI) algorithm of the SpikeCV (Zheng et al., 2026) library to reconstruct grayscale images from spike streams with dimensions $[T, H, W]$. This method exploits the spatiotemporal sparsity and encoding capacity of spike streams to approximate the textural structure of conventional images.

The core idea behind TFI is that the temporal interval between adjacent spikes encodes local texture intensity: shorter intervals correspond to higher pixel activity and brighter intensity. The algorithm identifies the two nearest spike timestamps within a bounded temporal window ($\pm\Delta t$) for each pixel and computes the grayscale of the value pixel based on their interval duration.

### A.2.4 ADAPTIVE LEAKY INTEGRATE-AND-FIRE NEURON MODEL

Common RSNN models often overlook several biologically relevant dynamic processes, particularly those occurring over extended time scales. To address this, we incorporate neuronal adaptation into our RSNN design. Empirical studies, such as those in the Allen Brain Atlas, indicate that a significant proportion of excitatory neurons exhibit adaptation with varying time constants. Several approaches to fit models for adapting neurons to empirical data.

In this work, we adopt a simplified formulation. The membrane potential of each neuron is updated using the Exponential Euler Algorithm (Cachia, 2004). The firing threshold $v_t^{th}$ increases by a fixed value $\beta_t$ following each spike, while the peak voltage $v_t^{\mathbf{th}} = \beta_t \cdot v_{\text{peak}}$ remains constant. At a discrete time step of $\Delta t = 1$ ms, the neuronal membrane potential update rule is defined as:

$$h_t = \alpha_t \cdot v_{t-1} + (1 - \alpha_t) \cdot (W_{\text{rec}} s_{t-1} + W_f s_t^{(l-1)}), \tag{6}$$

where $\alpha_t = \exp(-\Delta t / \tau_a)$ and $s \in \{0, 1\}$ denotes the binary spike train of neuron. The retention of membrane potential is controlled by a decay factor $\gamma_t$. In summary, the neuron model incorporates three adaptive parameters that govern the decay of the membrane potential, the firing threshold, and the reset potential, jointly determining the neuron's adaptive dynamics, respectively.

### A.2.5 Applying BPTT to RSNN

Although backpropagation through time (BPTT) is not biologically plausible, it serves as an effective optimization strategy for training RSNNs over extended temporal sequences, analogous to how evolutionary and developmental processes adapt biological systems to specific tasks. Prior work has applied backpropagation to feedforward spiking neural networks by introducing a pseudo-derivative to approximate the non-existent gradient at spike times. This surrogate derivative increases smoothly from 0 to 1 and then decays, allowing gradient-based learning. In our implementation, the pseudo-derivative's amplitude is attenuated by a factor less than 1, which improves BPTT stability and performance when training RSNNs over longer time horizons. Throughout, we use the following surrogate function $g(x)$:

$$g(x) = \text{sigmoid}(\alpha x) = \frac{1}{1 + e^{-\alpha x}}, \tag{7}$$

the spiking function of the sigmoid gradient is used in backpropagation. The gradient is defined by:

$$g'(x) = \alpha * (1 - \text{sigmoid}(\alpha x)) * \text{sigmoid}(\alpha x), \tag{8}$$

where $\alpha$ is a parameter that controls the slope of the surrogate derivative. Unless otherwise mentioned, we set $\alpha = 4$.

### A.2.6 Detail of Loss Function

In our training framework, the total loss $L$ is composed of three distinct components, each serving a specific purpose in optimizing the performance of the model. Specifically, the total loss is defined as

$$L = L_{\text{stereo}} + \lambda_f L_{\text{rate\_reg}} + \lambda_v L_{\text{v\_reg}}, \tag{9}$$

where $L_{\text{stereo}}$ represents stereo loss, $L_{\text{rate\_reg}}$ denotes the firing rate regularization loss, and $L_{\text{v\_reg}}$ corresponds to the velocity regularization loss. The parameters $\lambda_f$ and $\lambda_v$ are regularization coefficients that control the influence of the respective regularization terms. Stereo loss $L_{\text{stereo}}$ is designed to measure the discrepancy between the ground-truth disparity $d_{\text{gt}}$ and the predicted disparity $d_t$ over time $t$. It is formulated as follows:

$$L_{\text{stereo}} = \sum_{t=1}^{T} \eta^{T-t} ||d_{\text{gt}} - d_t||_1, \tag{10}$$

where $\eta = 0.9$ is a discount factor that assigns more weight to recent disparities, ensuring that the model focuses more on recent predictions.

The firing rate regularization loss $L_{\text{rate\_reg}}$ aims to stabilize the firing rates of neurons by penalizing deviations from the targe firing rate $r_0$. It is defined as:

$$L_{\text{rate\_reg}} = \sum_{i=1}^{N} (r_i - r_0)^2, \quad r_i = \frac{1}{T} \sum_{t=1}^{T} S_i(t), \tag{11}$$

where $r_i$ is the average firing rate of the neuron $i$ over time $T$, this term encourages the model to maintain consistent firing rates across neurons, thereby enhancing stability and preventing over-activation.

The voltage regularization loss $L_{\text{v\_reg}}$ is introduced to regularize the temporal dynamics of the model by penalizing the high velocities of the neurons. It is expressed as

$$L_{\text{v\_reg}} = \sum_{i=1}^{N} \sum_{t=1}^{T} v_i(t)^2, \tag{12}$$

where $v_i(t)$ represents the membrane potential of neuron $i$ at time $t$, and $N$ is the number of neurons in the network. This regularization term helps smooth the temporal behavior of the model, ensuring that changes in neuronal states are gradual and controlled.

In general, the combination of these loss components allows for a comprehensive optimization strategy that balances accuracy, firing rate stability, and temporal smoothness, thereby enhancing the robustness and performance of the model in stereo vision tasks.

### A.2.7 Network Re-training Method

When frame- or event- based networks are re-training, we make certain adjustments to the hyper-parameters of these baseline models. Firstly, we did not simply use the default hyperparameters from the original papers because frame-based and event-based methods differ significantly from spike-based data, and using default settings would clearly disadvantage some baselines. There-fore, every baseline was re-trained with targeted hyperparameter adaptation for the spike modality. Secondly, we did not perform an exhaustive full hyperparameter search for every baseline. Many baselines (e.g., Selective-Stereo, DLNR, RAFT-Stereo, ZEST) have large and complex hyperparam-eter spaces. Running a full grid or Bayesian search for each would require tens of thousands of GPU hours, far beyond the feasibility of this work. Thus, we adopt a practical, fairness-oriented tuning protocol. For each baseline, we tune the hyperparameters that are known to be sensitive to data modality, including:

**Frame-/Event-based baselines:**

- learning rate ($\pm 2\times$ around original)
- batch size (to account for temporal dimension, 4–16)
- temporal aggregation/voxelization hyperparameters
- photometric/contrastive loss weights where applicable
- number of update iterations (if models use iteration refinement, 16)
- correlation pyramid resolution
- random input crops used during training
- weight decay in optimizer ($10^{-5}$)

**Event-based models:**

- time-bin size
- event voxel grid normalization
- temporal decay parameters

We select the best settings on the synthetic validation dataset, and then retrain using the combined training set.

In summery, our objective was to give each baseline a reasonable and modality-aware advantage, rather than force them to operate with incompatible default settings. All baselines:

- ✓Retrain from scratch on the spike stereo dataset
- ✓Use modality-adjusted hyperparameters
- ✓Evaluate under the same data splits and metrics

This ensures a fair comparison, even though a full hyperparameter search is not computationally feasible.

### A.3 Details of the Synthetic Dataset

A conventional active stereo depth system typically comprises an infrared (IR) projector, a pair of IR stereo cameras (left and right) and a color camera. Depth measurement is achieved by projecting a dense IR dot pattern onto the scene, which is subsequently captured by the stereo IR cameras from different viewpoints. The depth values are then estimated by applying a stereo matching algorithm to the captured image pair, leveraging the disparity between them. We use the method from (Dai et al., 2022; Zhang et al., 2024a) that replicates this process, containing light pattern projection, capture, and stereo matching. The simulator is implemented based on the Blender rendering engine.

Table 5: Statistics of the training and test set of synthetic dataset.

|  | Scene Indexes | Number of Depth Map | Resolution | Number of Spike Frames | Number of Scenes |
|---|---|---|---|---|---|
| Training Dataset | 0 – 149 | 38,250 | $400 \times 250$ | 76,500 | 150 |
| Test Dataset | 150 – 189 | 10,200 | $400 \times 250$ | 20,400 | 40 |

### A.3.1 RANDOM SCENES

For each scene, we generate the corresponding '.meta' files to store structured metadata information. Additionally, for each object instance, a high-quality '.obj' mesh file is created to represent its geometry. To simulate realistic camera motion, a continuous trajectory is generated by sampling camera poses uniformly from a virtual sphere with a fixed radius of 0.6 meters. The representative camera viewpoints are separated by a constant angular increase of 3 °, ensuring smooth transitions along the trajectory. The objects are rotated accordingly to maintain consistent spatial alignment. Furthermore, a uniform background and consistent lighting configuration are applied across all scenes to reduce domain-specific biases and enhance rendering consistency. To further bridge the domain gap between synthetic and real-world data, we adopt domain randomization strategies by randomizing object textures, material properties (from specular, transparent, to diffuse), object layouts, floor textures, and lighting conditions aligned with camera poses.

### A.3.2 STEREO MATCHING

We perform stereo matching to obtain the disparity map, which can be transferred to the depth map leveraging the intrinsic parameters of the depth sensor. Specifically, a matching cost volume is constructed along the epipolar lines between the left and right infrared (IR) images. The disparity at each pixel is determined by selecting the position with the minimum matching cost. To enhance accuracy, sub-pixel disparity refinement is performed via quadratic curve fitting. In addition, several post-processing techniques are employed to improve depth quality, including left-right consistency checks, uniqueness constraints, and median filtering. These steps collectively yield more realistic and robust depth estimations. The calculation method from disparity to depth is as follows:

$$Z = \frac{Bf}{d + (c_{x1} - c_{x0})}, \tag{13}$$

where $Z$ is the depth obtained, $d$ is the disparity, $f$ denotes the focal length of the camera, $B$ is the baseline distance between the stereo cameras and $c_{x1} - c_{x0}$ represents the horizontal offset between the principal points of the right and left cameras.

### A.3.3 SYNTHETIC DATASET

We make use of domain randomization and depth simulation and construct a large-scale synthetic dataset. In total, the dataset consists of two subsets: (1) Training set: Consists of 76,500 spike streams and 38,250 depth maps generated from 150 scenes composed of 88 distinct objects, with randomized specular, transparent, and diffuse materials. (2) Test set: 20,400 spike streams and 10,200 depth maps generated from 40 new scenes containing 88 distinct objects. The scenes in the dataset are shown in Fig. 8 and Fig. 9, and the detailed statistics of dataset are shown in Table 5.

### A.4 DETAILS OF THE REAL DATASET

### A.4.1 STEREO CAMERA SYSTEM

We constructed a stereo imaging system using two spike cameras and one RGB-D camera (Kinect) to simultaneously capture left/right spike streams and the corresponding ground-truth depth images. As illustrated in Fig. 10, the two spike cameras are mounted on a high-precision alignment rig to ensure accurate baseline calibration. The spatial resolution of the spike camera is $400 \times 250$, and the temporal resolution is 20,000 Hz. The baseline between two spike cameras is 0.08 m, with a focal length of 16 mm for both. The resolution of the RGB-D camera is $1280 \times 720$, and the frequency is 30 FPS. Compared with LiDAR, the RGB-D sensor provides dense depth annotations within its

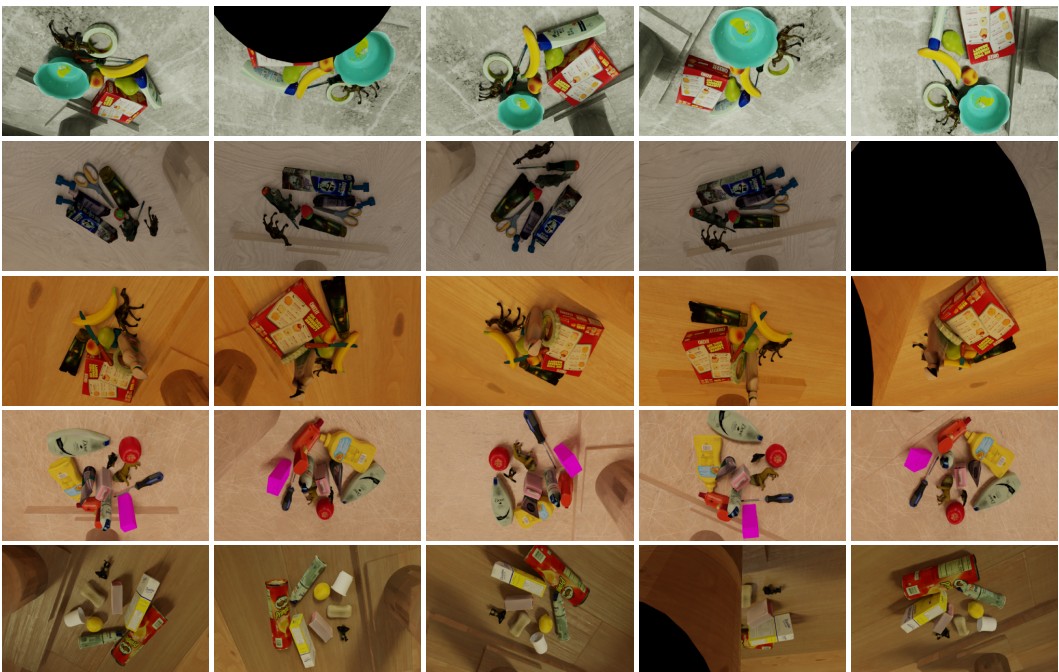

Figure 8: Scenes for generating the training set of the synthetic dataset. Selected five scenes, each displayed as a row in which the images correspond to different camera viewpoints of the same scene (partial views).

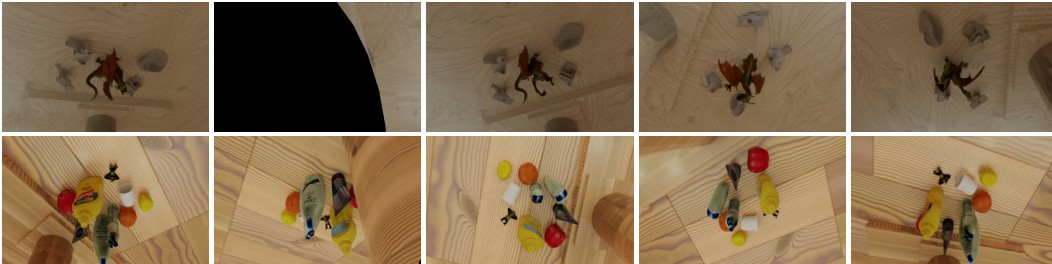

Figure 9: Scenes for generating the test set of the synthetic dataset. Selected two scenes, each displayed as a row in which the images correspond to different camera viewpoints of the same scene (partial views).

effective range, avoiding the sparse and incomplete depth output typically associated. To ensure spatial alignment between modalities, intrinsic and extrinsic calibrations are performed between the spike cameras and the depth camera. The captured depth maps are then geometrically warped to the coordinate frame of the left spike camera for accurate correspondence.

### A.4.2    STEREO SYSTEM CALIBRATION

For the calibration of spike cameras and Kinect, we used the following methods and programs. When calibrating with spike cameras, it is necessary to capture images of a standard chessboard calibration pattern. Approximately 20 sets of images should be taken from different positions and angles (covering left-right and up-down tilts of 15-45 degrees, various distances, and the entire field of view). During shooting, the calibration board must remain stationary, be fully visible in the field of view of both spike cameras, and both cameras should capture images simultaneously. Sufficient time should be allowed at each position to ensure image clarity. Additionally, when reconstructing images using TFI/TFP (Zheng et al., 2026), the hardware and software settings (including the TFI/TFP parameters) of the two spike cameras must be kept consistent. To reduce noise interfer-

ence, the process should start with completely static scenes and longer TFI windows, ensuring that the scenes captured by both cameras are identical except for perspective differences. Adequate lighting should be provided, and scenes with clear, and strong edges should be selected to improve calibration success rates. OpenCV's calibration algorithms are used for stereo rectification (the primary APIs are "cv.stereoRectifyUncalibrate" and "cv.stereoRectify"). The rectification is applied to the TFI/TFP images reconstructed by the spike cameras, and the rectified images serve as input to the binocular stereo algorithm.

### A.4.3 STEREO SYSTEM SYNCHRONIZATION MECHANISM

We employ a hybrid hardware-trigger + software algorithmic synchronization pipeline that ensures stereo alignment suitable for downstream training and evaluation as follows:

**Hardware-level coarse synchronization.** Both spike cameras are connected to the same host machine via high-speed USB. During recording, we use a shared software trigger to initiate acquisition on both sensors: $t_0^{(L)} \approx t_0^{(R)}$, where $t_0^{(L)}$ and $t_0^{(R)}$ are the left/right start timestamps. This provides coarse alignment, typically within tens of microseconds.

**Empirical estimation of residual time offsets** Even with shared triggering, small timing mismatches remain due to independent internal clocks. To measure the remaining timing error, we place a fast-switching LED in both cameras' fields of view and compare the spike timestamps of each illumination transition. We collect over 2,000 pairs of real stereo sequences and estimate the residual time offsets:

$$\Delta t = t_i^{(L)} - t_i^{(R)}, \qquad i = 1, \dots, N. \tag{14}$$

We model the empirical distribution as:

$$\Delta t \sim \mathcal{N}(\mu_{\Delta t}, \sigma_{\Delta t}^2), \tag{15}$$

where in practice: $\mu_{\Delta t}$ is within 3–7 $\mu s$,

$\sigma_{\Delta t}$ depends on lighting and spike density but remains ¡ 15 $\mu s$.

This analysis characterizes the realistic synchronization noise between the two spike streams.

**Temporal cost matching.** Before building the cost volume, we apply a lightweight alignment procedure:

$$S^{(L)}(t) \rightarrow S^{(L)}(t + \hat{\Delta}t), \tag{16}$$

where $\hat{\Delta}t$ is estimated via maximizing temporal correlation:

$$\hat{\Delta}t = \arg\max_{\delta} \sum_{x,y} \langle S^{(L)}(x, y, t + \delta), S^{(R)}(x, y, t) \rangle. \tag{17}$$

This provides fine temporal adjustment without modifying the raw spike stream statistics.

**Domain Randomization in training.** To ensure robustness to synchronization jitter, we inject the measured $\Delta t$ distribution into the synthetic spike generator:

$$S^{(L)} * \text{syn}(t) = S^{(L)} * \text{ideal}(t + \epsilon), \quad \epsilon \sim \mathcal{N}(\mu_{\Delta t}, \sigma_{\Delta t}^2). \tag{18}$$

This domain randomization teaches SpikeStereoNet to become invariant to microsecond-level offsets.

**Fine-tuning on real spike data.** Because the network is pre-trained with jitter matching the real distribution, fine-tuning on real data automatically adapts to the actual synchronization statistics:

$$\epsilon_{\text{real}} \in \text{support}(\mathcal{N}(\mu_{\Delta t}, \sigma_{\Delta t}^2)). \tag{19}$$

As a result, the RSNN refinement module effectively compensates for minor mismatches during iterative updates.

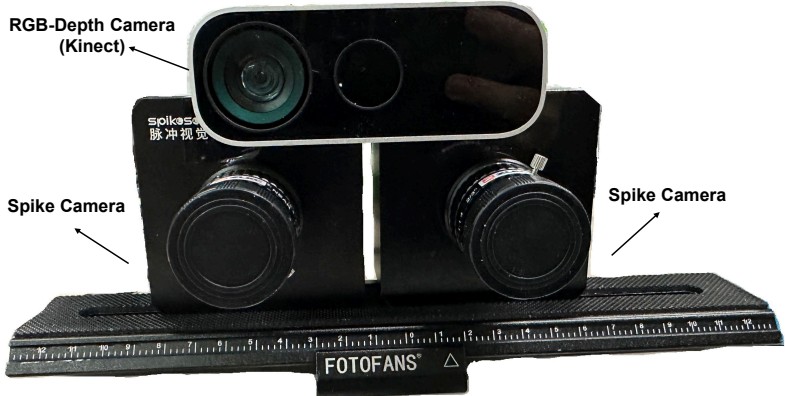

Figure 10: The photograph of our hybrid imaging system. The RGB-D camera is fixed on the top plane of the stereo spike cameras. The extrinsic parameters between the cameras are calibrated.

### A.4.4 REAL DATASET COLLECTION

The real dataset comprises some video sequences captured under diverse environmental conditions, including varying ambient illumination levels (e.g., low, medium, and high brightness) and a range of dynamic scenarios characterized by distinct object arrangements. These scenes encompass different spatial configurations and relative motions of multiple objects (e.g., camera shake and object motion), introducing varied degrees of motion blur. The following is the specific collection process of our real dataset:

**Dataset Acquisition Setup.** Our real dataset is collected using a custom-built hardware system consisting of:

- Two spike cameras arranged in a stereo configuration, with a baseline of 80 mm (calibrated for epipolar alignment).

- A Kinect depth camera through hardware triggering, used to capture ground-truth depth for alignment.

**Calibration of the Stereo Spike Camera System.** Stereo spike cameras are calibrated following a rigorous protocol:

- A matte chessboard calibration target ($10\times14$ squares, 25 mm per square) is used, positioned statically across 20+ viewpoints (covering 15-45° tilts and 0.5-2 m distances).

- Both spike cameras captured synchronized spike streams of the target, which are converted to TFI (Temporal Filtered Integration) images with a 50 ms window to ensure clarity.

- The calibration quality is validated by checking that the reprojection errors for the 3D chessboard corners are $< 1$ pixel across all viewpoints.

**Dataset Content.** The dataset includes indoor real-world scenes:

- Objects: 20+ everyday items, including textureless objects (e.g., plain plastic basin, white mugs) and reflective objects (e.g., metal box). Each scene contains 5-8 randomly arranged objects to simulate clutter.

- Lighting Conditions: 8 controlled lighting setups with direct light sources (LED panels) at angles of $30°, 45°, 60°$, and $90°$ relative to the scene, with intensities ranging from 300-1500 lux to induce varying levels of specularity and shadow.

- Data Modalities: For each scene, we provide raw stereo spike streams and Kinect-aligned depth maps (ground truth).

### A.5 MORE EXPERIMENTAL RESULTS

#### A.5.1 THEORETICAL ANALYSIS OF NETWORKS

We further supplement the theoretical analysis in the article with the explicit convergence of RSNN and Lipschitz stability analysis. Here is the complete proof:

**Constructing Fixed Point Iteration and Mapping**. For clarity, we rewrite the RSNN update step in vector form:

$$\mathbf{u}_t = F(\mathbf{u}_{t-1}) \quad \text{with} \quad \mathbf{u} := (\mathbf{h}, \mathbf{v}, \mathbf{s}, \mathbf{v}^{\text{th}}) \in \mathbb{R}^{4N}, \tag{20}$$

$$F(\mathbf{u}) = \begin{bmatrix} \alpha_t \mathbf{v} + (1 - \alpha_t)\big(W_{\text{rec}}\theta(\mathbf{h} - \mathbf{v}^{\text{th}}) + W_f \mathbf{s}^{(l-1)}\big) \\ \beta_t v_{\text{peak}} \\ \theta(\mathbf{h} - \mathbf{v}^{\text{th}}) \\ \mathbf{h} - \gamma_t \theta(\mathbf{h} - \mathbf{v}^{\text{th}})\mathbf{v}^{\text{th}} \end{bmatrix}. \tag{21}$$

The parameters $\alpha_t, \beta_t, \gamma_t$ are all adaptive variables regulated by the firing rate, and after the activation function, their ranges are all $(0, 1)$, that is, $0 < \alpha_t < 1, 0 < \beta_t < 1, 0 < \gamma_t < 1$.

**Lipschitz Continuity.** The Lipschitz constant is determined by the activation function $\theta(\cdot)$ (hard threshold/ReLU) and a linear operator. If $\theta$ is 1-Lipschitz ReLU, the mapping $F$ is globally Lipschitz in $\mathbf{u}_t$, then:

$$\|F(\mathbf{u}) - F(\mathbf{u}')\| \le L\|\mathbf{u} - \mathbf{u}'\|, \tag{22}$$

where

$$L = \alpha_t + (1 - \alpha_t)\,\gamma_t\,\beta_t\,v_{\text{peak}}\,\|W_{\text{rec}}\|, \tag{23}$$

and all the ranges of variables in this equation are $(0, 1)$, hence the RSNN update is a contraction with constant $L < 1$.

**Iterative Refinement and Convergence.** Because $F$ is a contraction in the entire metric space $(\mathbb{R}^N \|\cdot\|)$, and according to the Banach Fixed-Point theorem (Gordji et al., 2017), the error satisfies.

$$\|\mathbf{u}^{(k)} - \mathbf{u}^*\| \le \frac{L^k}{1 - L}\|\mathbf{u}^{(1)} - \mathbf{u}^{(0)}\|. \tag{24}$$

Hence, the sequence $\{\mathbf{u}_t\}$ linear exponentially to a unique fixed point $\mathbf{u}^\star$. The bound agrees with the empirical spectral-radius curves reported earlier (Fig. 7).

**Practical Implications.**

- No Exploding States: The membrane potentials remain bounded by $\frac{1}{1-L}\|\mathbf{u}^{(0)}\|$.

- Training Stability: Gradients passed through time have an upper bound $L^{T-t}$, which mitigates the exploding BPTT signals.

- Noise Resilience: Any perturbation $\delta$ at step $k$ decays as $L^{t-k}\delta$.

Under the condition that $\alpha_t, \beta_t, \gamma_t$ are bounded and the weight norm satisfies $L < 1$, the iterative $\mathbf{u}_t = F(\mathbf{u}_{t-1})$ of RSNN linearly converges to a unique fixed point, providing a theoretical guarantee for continuous and contraction Lipschitz mapping. The above theoretical proof supplements the empirical stability study already proposed (Fig. 7).

#### A.5.2 ADDITIONAL COMPARATIVE RESULTS

We present a complete stereo depth estimation sequence for a single scene within the synthetic dataset, capturing the full motion trajectory of objects throughout the dynamic environment. The different scene images below are obtained by rotating the stereo spike camera system from different views, and we have extracted some of the views at equal intervals for display. As illustrated in Fig. 11 and Fig. 12, the predicted depth maps effectively reconstruct the continuous evolution of the scene, accurately reflecting both spatial structure and temporal consistency. These results demonstrate the model's ability to preserve coherent geometry across time, validating its effectiveness in handling dynamic visual inputs. The supplementary materials in the zip file include continuously changing videos.

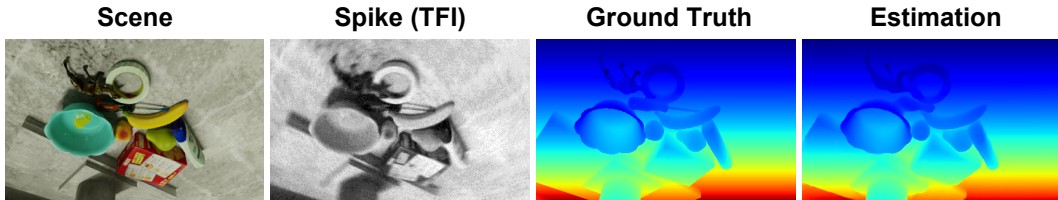

Figure 11: Video results of our method on the synthetic dataset in the single scene. From left to right, the figure shows the sequential scene images, the visualized spike stream using the TFI method, the ground-truth depth maps, and the depth estimation results produced by our proposed method.

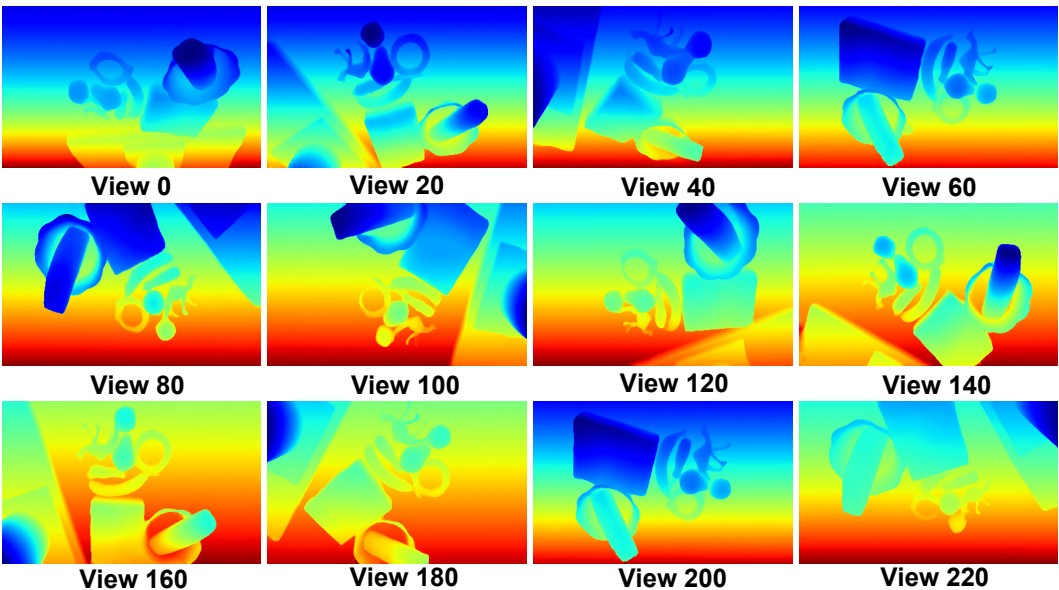

Figure 12: Visualization results of SpikeStereoNet on the synthetic dataset in a single scene. We presented depth estimation results of 12 views in the scene.

In addition, we conducted a qualitative comparison between SpikeStereoNet and several state-of-the-art methods on the synthetic dataset. Our method consistently produces more accurate and visually coherent depth maps in various scenes. Further qualitative results are provided to demonstrate the robustness and effectiveness of our approach.

### A.5.3 SENSITIVE OF THRESHOLD AND TEMPORAL QUANTIZATION

We conducted additional experiments to quantify how the spike threshold and temporal quantization affect SpikeStereoNet. The model shows robustness to a wide range of settings, largely due to the adaptive LIF dynamics and the RSNN refinement stage.

During spike generation, the threshold controls how many potentials should accumulate before a spike is triggered. We evaluate thresholds in the range: $v'_{\text{peak}} \in \{ 0.7v_{\text{peak}}, 1.0v_{\text{peak}}, 1.3v_{\text{peak}}, 1.6v_{\text{peak}} \}$, where $v_{\text{peak}}$ is the nominal threshold. Spike streams are discretized into $(T)$ bins: $\Delta t \in \{0.5 \text{ ms}, 1.0 \text{ ms}, 2.0 \text{ ms}, 4.0 \text{ ms}\}$, and $\Delta t = 1.0$ ms is the default time step of the model. The comparison results are shown in Table 6.

Overall, the impact of the two parameters on the model is as follows:

- Low threshold for denser spikes and slightly more noise.
- High threshold for sparser spikes and fewer temporal cues.
- Very fine bins ($< 0.5$ ms) increase sparsity but yield negligible improvement.

Table 6: Sensitivity analysis of the spike threshold and the temporal quantization step during spike generation.

| Settings | bad 2.0 (%) ↓ | bad 3.0 (%) ↓ | AvgErr (px) ↓ | Variation |
|---|---|---|---|---|
| $0.7v_{\text{peak}}$ | 4.56 | 2.61 | 0.44 | $+3.5\%$ |
| $1.3v_{\text{peak}}$ | 4.78 | 2.62 | 0.45 | $+5.9\%$ |
| $1.6v_{\text{peak}}$ | 4.92 | 2.78 | 0.46 | $+8.3\%$ |
| $v_{\text{peak}}, \Delta t = 1$ ms | **4.13** | **2.38** | **0.42** | – |
| $\Delta t = 0.5$ ms | 4.44 | 2.58 | 0.43 | $+1.2\%$ |
| $\Delta t = 2.0$ ms | 4.93 | 2.67 | 0.44 | $+3.6\%$ |
| $\Delta t = 4.0$ ms | 5.01 | 2.80 | 0.47 | $+10.7\%$ |

Table 7: The comparison results of the computational efficiency.

| Method | Equivalent FPS | FLOPs (B) | Method | Equivalent FPS | FLOPs (B) |
|---|---|---|---|---|---|
| CREStereo | 1.67 | 863 | GMStereo | 2.28 | 160 |
| RAFT-Stereo | 1.80 | 798 | DLNR | 1.79 | 1580 |
| ZEST | 1.63 | 989 | MoCha | 1.31 | 935 |
| IGEV-Stereo | 1.53 | 614 | MonSter | 1.72 | 1567 |
| Selective-Stereo | 1.76 | 957 | Ours | 1.91 | 473 |

- Coarse bins ($> 4$ ms) lose spike timing, which is important for handling motion.

- The performance of the model varies within $\pm 5.9\%$ thought the threshold range.

- Within 0.5–2 ms, the model is stable with the AvgErr variation of less than $3.6\%$.

This robustness arises because the dynamics of ALIF incorporates adaptive membrane thresholds $\beta_t v_{\text{peak}}$ and learned reset gates $\gamma_t$ that automatically compensate for moderate changes in spike density, even when the raw spike discretization changes.

SpikeStereoNet is not strongly sensitive to moderate shifts in threshold or temporal step because: The ALIF neuron's adaptive threshold dynamically regulates firing rates. The RSNN recurrence reconstructs the missing temporal structure from the context. Firing-rate and voltage regularization constrain membrane dynamics, ensuring stable behavior even under different spike densities. Overall, only extreme settings (very coarse temporal step or high thresholds) noticeably degrade performance.

### A.5.4 COMPUTATIONAL EFFICIENCY

We conducted additional experiments to measure the FPS or Inference Latency of the networks listed in Table 1 using the same hardware configuration: a NVIDIA RTX 3090 GPU, paired with an Intel Core i7-13700K CPU and 32GB RAM. All models are averaged over 100 inference runs to minimize variability. The results are summarized, the FPS and FLOPs of inference refer to a depth map estimated using a set of stereo spike streams ($T = 50$).

From the additional results in the Table 7, it can be seen that our model also outperforms most in inference speed and FLOPs.

### A.5.5 HYBRID MODALITY FUSION

Hybrid neuromorphic-frame sensing is becoming increasingly common, and we have conducted preliminary hybrid experiments and discussed integration strategies below.

**Spike+TFI Image Fusion.** We generated TFI images (Temporal Frame Integration, 10–50 windows) from the raw spike stream:

$$\text{TFI}(x, y) = \sum_{t=t_0}^{t_0+\Delta t} S(x, y, t). \tag{25}$$

Table 8: The quantitative results of Spike+TFI fusion on the synthetic dataset.

| Method | bad 2.0 (%) ↓ | bad 3.0 (%) ↓ | AvgErr (px) ↓ |
|---|---|---|---|
| Spike+TFI ($\Delta t = 10$) | 6.18 | 3.69 | 0.67 |
| Spike+TFI ($\Delta t = 30$) | 5.35 | 3.09 | 0.56 |
| Spike+TFI ($\Delta t = 50$) | 4.98 | 2.87 | 0.54 |
| **Ours** | **4.13** | **2.38** | **0.42** |

Table 9: The quantitative results of Spike+RGB fusion on the real dataset.

| Method | bad 2.0 (%) ↓ | bad 3.0 (%) ↓ | AvgErr (px) ↓ |
|---|---|---|---|
| Spike+RGB (Kinect) | **5.08** | 3.43 | 0.59 |
| Ours | 5.33 | **3.19** | **0.56** |

To evaluate fusion, we adapted our architecture by adding a TFI branch and merging its features with spike features via: channel-wise concatenation, and context fusion (applying a shallow CNN to extract context features before cost-volume refinement). The experimental results on synthetic datasets are as Table 8.

Compared to only spike input, TFI provides limited benefits in motion situations where spike time is crucial. Quantitative evaluation shows that performance has decreased by about 12% in average scenes. Because using TFI for reconstruction into frame images additionally introduces noise.

**Experiments With Spike+RGB Frames.** We also performed preliminary fusion with synchronous RGB frames from the Kinect. Following prior hybrid designs, we evaluated two fusion strategies. (1) Early Fusion. The RGB data are downsampled and concatenated with the spike feature map, which has the advantages of simplicity and disadvantages of misalignment and HDR mismatch. (2) context Fusion. RGB is processed by a separate context encoder, producing $F_{rgb} = \phi_{rgb}(RGB)$, $F_{spike} = \phi_{spike}(S)$, followed by $F_{fused} = \text{Conv}([F_{rgb}, F_{spike}])$. This version is consistently more stable. The experimental results on real datasets are as Table 9. In summary, spike+RGB yields an improvement in object details, with slightly better global structure consistency, but with small degradation in motion (RGB lagging behind spikes). In addition, due to the use of Kinect depth cameras to capture RGB images, frame rate and resolution blur output cannot be avoided.

In summary, we conducted preliminary spike+TFI and spike+RGB fusion experiments. The hybrid inputs offer modest robustness improvements in static illumination scenes. We describe a preliminary architecture modification for hybrid fusion, inspired by the hybrid spike-RGB literature (Chang et al., 2023).

### A.5.6 ADDITIONAL RESULTS FOR ABLATION STUDY

We provide additional ablation studies to demonstrate the effectiveness of our designed modules.

**RSNN Module.** Below we disentangle the impact of group normalization (GN) from that of our RSNN refinement by re-running the key variants and reporting the results in the same metrics used in Table 10. The RNN is the same number of neurons as the original model.

These results demonstrate that both components, the RSNN architecture and the GN contribute to the performance, but neither alone is sufficient to achieve our full model's results. Thus, the competitive performance of our framework arises from the synergy between the RSNN (optimized for spike-based temporal patterns) and the GN (stabilizing its dynamics).

**Adaptation of Neuron Models.** We conducted an explicit replacement experiment in which the adaptive LIF (ALIF) neurons used in our recurrent refinement were substituted with a classical LIF or raw SNN implementation that keeps all adaptive scalars fixed. In the following, we formalize the LIF classical, highlight the theoretical differences, and report quantitative results.

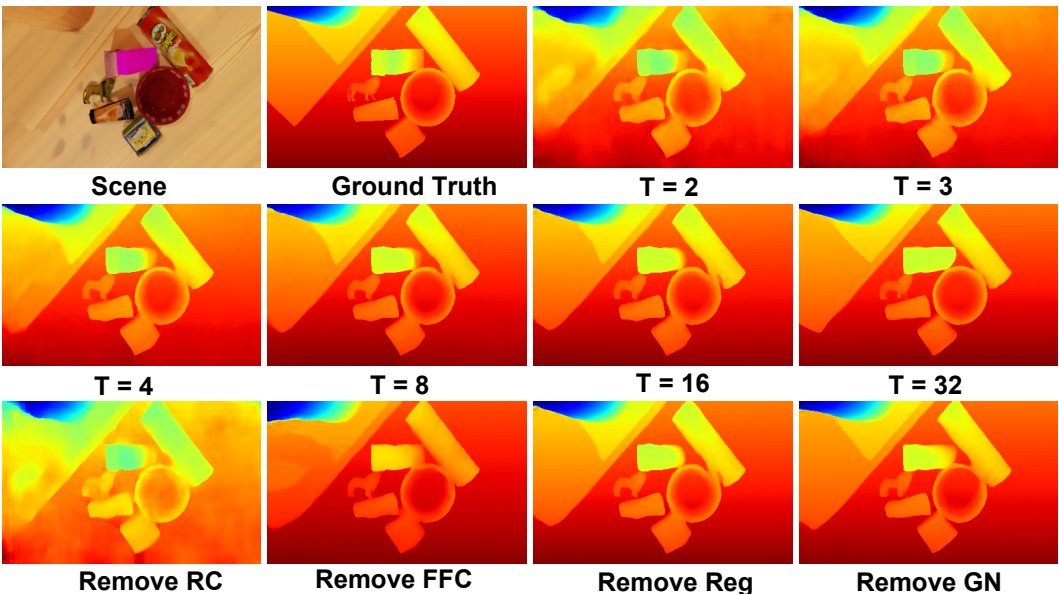

Figure 13: Visualization results of ablation study for evaluating the effectiveness of our framework on the synthetic dataset. Among them, T = 32 is the baseline of our model.

The raw SNN layer with no recurrent connections and fixed threshold (feedforward spiking layer) typically contains:

$$
\begin{aligned}
v_t &= \alpha v_{t-1} + W_f s_t^{(l-1)}, \\
s_t &= \theta(v_t - v_{\text{th}}).
\end{aligned}
\tag{26}
$$

The classical LIF neuron, unlike our ALIF, uses fixed parameters (no adaptive variables $\boldsymbol{\alpha_t}, \boldsymbol{\beta_t}, \boldsymbol{\gamma_t}$). Its dynamics can be written as:

$$
\begin{aligned}
\boldsymbol{h_t} &= \alpha \cdot \boldsymbol{v_{t-1}} + (1 - \alpha) \cdot (W_{\text{rec}} \boldsymbol{s_{t-1}} + W_f \boldsymbol{s_t^{(l-1)}}), \\
v^{\text{th}} &= \beta \cdot v_{\text{peak}}, \\
\boldsymbol{s_t} &= \theta(\boldsymbol{h_t} - v^{\text{th}}), \\
\boldsymbol{v_t} &= \boldsymbol{h_t} - \gamma_t \cdot \boldsymbol{s_t} \cdot v^{\text{th}},
\end{aligned}
\tag{27}
$$

where $\alpha$, $\beta$ and $\gamma$ are fixed constants (e.g., 0.5, 1.0, 1.0 in our baselines), not learnable or input dependent. This removes adaptivity to temporal patterns in spike streams. All other network components, learning rates, and losses were kept identical. We conducted an ablation study by replacing ALIF with classical LIF or raw SNN in the synthetic dataset (the same architecture). All other conditions of the experiment are the same, and the results are in Table 4.

From the above table, it can be seen that the overall performance decreases significantly after replacement with the LIF model. The ALIF is critical for our task because stereo spike streams require adaptive temporal processing. Adaptive gating markedly improves both stability and final accuracy with negligible overhead, justifying our choice of the novel ALIF over the classical LIF or raw SNN.

**Number of Iterations.** We analyze the impact of iteration steps by varying the number of iterations T in our RSNN-based model. As shown in Table 11, our method achieves competitive or superior performance with significantly fewer iterations compared to existing approaches. Unlike conventional GRUs or transformer-based update modules, RSNNs require fewer parameters and operations per iteration, thus maintaining a lower computational cost.

In addition, to further support the ablation studies presented in the main experimental section, we provide the corresponding qualitative results in Fig. 13. These visualizations clearly demonstrate the effectiveness of each module in our proposed framework and validate their individual contributions to the overall performance.

Table 10: Ablation study for RSNN and group normalization.

| Method | bad 2.0 (%) ↓ | bad 3.0 (%) ↓ | AvgErr (px) ↓ |
|---|---|---|---|
| RNN | 7.88 | 4.04 | 0.76 |
| RNN + GN | 7.28 | 3.41 | 0.66 |
| RSNN | 6.38 | 2.85 | 0.51 |
| **RSNN + GN (Ours)** | **4.13** | **2.38** | **0.42** |

Table 11: Ablation study for number of iterations and runtime.

| | Number of Iterations (T) | | | | | |
|---|---|---|---|---|---|---|
| | 2 | 3 | 4 | 8 | 16 | 32 |
| AvgErr (px) | 0.69 | 0.61 | 0.56 | 0.49 | 0.46 | 0.42 |
| Runtime (s) | 0.41 | 0.42 | 0.43 | 0.46 | 0.51 | 0.63 |

**Image to Video.** Regarding our model for generating continuous spike streams using the video frame interpolation method. Below we justify our design choice, quantify the interpolation error, and report a new control experiment based on Blender's native sub-frame rendering and optical flow supervision.

We use EMA-VFI (Enhanced Motion-Aware Video Frame Interpolation) (Zhang et al., 2023), a successor to FILM that achieves comparable or superior quality in small motion scenes. In our raw Blender renders (ground truth), the quantitative metrics of EMA-VFI (PSNR = 38.2±1.5 dB, SSIM = 0.97±0.02) indicate minimal artifacts, with visual inspections confirming that interpolated frames preserve fine details critical for spike simulation. We computed the optical flow magnitude per pixel (in pixels) across the entire synthetic dataset.

From Table 12, more than 95% of the frames exhibit sub-pixel motion, far below the motion threshold ($\approx 4$ px) at which EMA-VFI begins to introduce visible artifacts. Thus interpolation artifacts are negligible for the vast majority of the data.

After interpolation, frames are passed through the spike simulator. The conversion from interpolated frames to spike streams via the spike simulator further mitigates residual artifacts. The simulator introduces Poisson-like temporal noise and sparsity, which naturally smooths minor interpolation inconsistencies. Thus, even if an interpolated frame has minor inaccuracies, the temporal difference between two adjacent bins remains dominated by photon shot noise injected by the simulator. Regarding ground-truth depth labels, which are extracted directly from Blender's z-buffer per original key-frame, and they do not rely on the interpolated RGB images. During spike simulation, the disparity is linearly blended between key-frames, guaranteeing label consistency regardless of interpolation artifacts.

To compare the effectiveness of the methods, we re-rendered full sequences with ground-truth optical flow from Blender, and regenerated a "SubFrame-GT" variant. The ground-truth optical flow exported via Blender's Vector pass, generate image data and passed to the spike simulator. We re-train the model from scratch using the same hyper-parameters, and the results are as Table 13.

The performance difference is $< 2\%$ / 0.1 pp, indicating that interpolation artifacts do not materially influence training or final accuracy. Overall, the two methods represent different patterns and approaches that can be chosen for specific tasks and datasets.

Table 12: Motion statistics in the synthetic datasets.

| Quantile | 50% | 75% | 90% | 95% |
|---|---|---|---|---|
| Motion (px) | 0.31 | 0.57 | 1.12 | 1.89 |

Table 13: Ablation study for Video frame interpolation methods.

| Dataset variant | bad 2.0 (%) ↓ | AvgErr (px) ↓ |
|---|---|---|
| SubFrame-GT (no interpolation) | 0.44 | 4.43 |
| Original (Ours) | 0.44 | 4.52 |

## A.6 BROADER IMPACTS

The proposed framework for stereo depth estimation from spike streams has a strong potential to advance neuromorphic vision in diverse application scenarios. In autonomous driving, the enhanced depth estimation enabled by spike-based sensing can improve 3D scene understanding, obstacle detection, and object location, thus contributing to more robust and reliable navigation systems. In the field of robotics, the fine-grained depth perception afforded by this method supports precise manipulation, mapping, and motion planning, especially in highly dynamic or high-speed environments where conventional cameras typically underperform. Moreover, this framework that can directly estimate depth from raw stereo spike streams under supervised settings, enabling broader exploration of spike-based vision in both research and industrial applications.

**Limitations.** Our method is limited by the small scale of the real world dataset and the domain gap between the synthetic and real data. Although domain adaptation helps, its effectiveness is constrained by the quality of real labeled data and cannot fully bridge distribution differences. We plan to extend our method to handle these questions in future work.

**Future Works.** We will strengthen the benchmark with a broader coverage, and therefore we have launched a follow-up data collection campaign that will: (1) Expand the variety of the scene: indoor, outdoor and semi-outdoor environments with larger depth ranges and moving platforms; (2) Increase lighting diversity: day–night sweeps and high dynamic range sequences; (3) Enrich object categories: targeting $\geq 250$ physical objects in 30 WordNet classes, including specular and transparent materials. The existing and extended dataset will be released publicly under a permissive license and the accompanying capture/annotation pipeline will be open-sourced to facilitate community contributions.

