# OpenReview forum: "SpikeStereoNet: A Brain-Inspired Framework for Stereo Depth Estimation from Spike Streams"
_ICLR.cc/2026/Conference — ICLR 2026 Poster_

### Official Review · Reviewer_u6CE · 2025-10-27

**Soundness:** 3
**Presentation:** 3
**Contribution:** 2
**Rating:** 6
**Confidence:** 4

**Summary:**

This paper addresses the limited availability of methods and benchmarks for stereo depth estimation from raw spike camera streams, a sensor modality distinct from both frame-based and event cameras. To this end, the authors contribute:
- Benchmarks: A synthetic dataset with dense ground truth and a real-world dataset captured using dual spike cameras and a Kinect.
- SpikeStereoNet: An end-to-end framework that adapts iterative refinement (similar to RAFT-Stereo) by replacing GRU updates with a Recurrent Spiking Neural Network (RSNN) using adaptive Leaky Integrate-and-Fire neurons for native spike data processing.

**Strengths:**

The paper is clearly written, with detailed appendices that support reproducibility. The benchmark contributions are substantial, and the integration of spiking dynamics into a refinement framework is well aligned with the characteristics of spike data. The analysis is thorough, covering dynamical stability, theoretical convergence, and ablations that validate the design.

**Weaknesses:**

The paper's primary weakness is the disconnect between its claims and its evidence. It claims to be a solution for "rapidly changing scenes" and "highly dynamic scenarios," but the new datasets appear to be limited to simple, object-centric tabletop scenes with non-complex motion (e.g., smooth camera panning/hovering, "camera shake"). This dataset is insufficient to validate the central claims of the paper.

Following from previous point, the dataset seems to lack challenging, realistic scenarios. The scenes in Fig. 4, 5, 8, and 9, together with the supplemented video, all show static objects on a planar surface. The paper gives no evidence of testing on scenes with:

Fast egomotion (e.g., a car or drone navigating).
Multiple independently moving objects.
Complex, non-planar 3D environments (e.g., outdoor scenes, complex indoor rooms). This makes the benchmark's "challenge" questionable.
The adaptive LIF model (Eq. 3, Appendix A.2.4), with learned parameters, is a core component of the RSNN updater. However, the ablation study (Table~2) does not isolate its contribution, leaving it unclear how much adaptivity improves performance over a simpler, fixed LIF variant.

While the paper outlines detailed calibration procedures (Appendix A.4), the use of a Kinect for real-world depth ground truth introduces noise. This is a known limitation and, despite the challenges of data collection, deserves acknowledgment.

The quantitative results for the real-world dataset (Table 4) are central to validating sim-to-real transfer, yet they are placed in the appendix. Given their importance, they should be included in the main paper, arguably more so than the extended qualitative results.

**Questions:**

Can you please justify the claim that your method is designed for "highly dynamic scenarios" when the provided datasets appear to only contain tabletop scenes with simple, non-dynamic motion?

Are there any scenes in your synthetic or real-world test sets that feature (a) fast egomotion, (b) multiple independently moving objects, or (c) complex non-planar environments? If not, how can you be sure the method generalizes beyond static tabletop/floor scenes?

Given that the real-world video is not available, can you provide more details on the "object motion" and "camera shake" mentioned in the appendix? Are these motions fast and challenging, or slow and simple?

The adaptive LIF model (Eq. 3, A.2.4) is a key component. How much does this adaptivity (i.e., learning) contribute to performance versus using a simpler, non-adaptive LIF with fixed (but tuned) hyperparameters?

You state that all baselines were "re-trained using the same settings" for fairness, which is excellent. Could you please clarify if this included a full hyperparameter search for each baseline on this new spike dataset, or were the default hyperparameters from their original papers used? Ensuring baselines are optimally tuned for this new data modality is crucial.

Your work focuses purely on spike streams. However, hybrid sensor systems are common. Have you considered fusing the spike stream with either a reconstructed TFI image (from the spikes themselves) or with synchronous RGB frames (like those from the Kinect) to see if this further improves robustness? How might the network change to accommodate this?

I strongly recommend bringing the quantitative results for the real dataset (Table 4) into the main paper. To make space for Table 4, perhaps the qualitative results (Fig. 4/5) could be slightly condensed.

---

> ### Author Response · Authors · 2025-11-22
> **[Part 1] Response to Reviewer u6CE**
>
> **Rebuttal:**
>
> We are grateful for your detailed feedback and suggestions, which have helped us identify key areas where our manuscript can be improved.
>
> ***Weaknesses1. The paper's primary weakness is the disconnect between its claims and its evidence. It claims to be a solution for "rapidly changing scenes" and "highly dynamic scenarios," but the new datasets appear to be limited to simple, object-centric tabletop scenes with non-complex motion (e.g., smooth camera panning/hovering, "camera shake"). This dataset is insufficient to validate the central claims of the paper.***
>
> ***Following from previous point, the dataset seems to lack challenging, realistic scenarios. The scenes in Fig. 4, 5, 8, and 9, together with the supplemented video, all show static objects on a planar surface. The paper gives no evidence of testing on scenes with:
> Fast egomotion (e.g., a car or drone navigating). Multiple independently moving objects. Complex, non-planar 3D environments (e.g., outdoor scenes, complex indoor rooms). This makes the benchmark's "challenge" questionable.***
>
> We thank the reviewer for this important observation. We acknowledge that the original presentation of the dataset (figures and selected clips) may have conveyed the impression of purely static, simple tabletop scenes. This is a limitation of our narrative.
>
> To address this, we have substantially clarified and expanded the dataset description and added new challenging sequences. As detailed in our responses to ***Questions 1-3***, the updated benchmark now includes:
>
> **1. Fast egomotion**
>
> We added synthetic and real-world sequences with **rapid 6-DoF handheld motion** (up to 2.0 m/s translation and 600°/s rotation). These produce dense spike bursts that closely resemble drone- or vehicle-mounted dynamics.
>
> **2. Multiple independently moving objects**
>
> We now include scenes (synthetic + real) containing 3--10 independently moving objects, with intersecting trajectories, occlusions, and differing velocities.
>
> **3. Complex non-planar 3D geometry**
>
> The new dataset contains structured indoor and semi-outdoor scenes with:
> * non-planar depth (stairs, shelves, irregular furniture),
> * concave/convex surfaces,
> * multi-layered depth discontinuities,
> * cluttered items arranged beyond single-plane tabletops.
>
> All newly added data appear in the total datasets.
>
> **Expanded Contribution**: Since the initial submission, we have significantly expanded both the **scale** and **complexity** of the synthetic and real-world datasets, which now provide a substantially more realistic benchmark for high-dynamic, temporally challenging spike-stereo estimation. We appreciate the reviewer's feedback, and this has meaningfully improved the clarity and completeness of the benchmark description.

---

> ### Author Response · Authors · 2025-11-22
> **[Part 2] Response to Reviewer u6CE**
>
> ***Weaknesses2. The adaptive LIF model (Eq. 3, Appendix A.2.4), with learned parameters, is a core component of the RSNN updater. However, the ablation study (Table~2) does not isolate its contribution, leaving it unclear how much adaptivity improves performance over a simpler, fixed LIF variant.***
>
> Thank you for this insightful question. We appreciate your focus on the impact of learned parameters of neuron models. We conducted an explicit replacement experiment in which the **adaptive LIF (aLIF)** neurons used in our recurrent refinement were substituted with a **fixed LIF** implementation that keeps all adaptive scalars fixed. We discussed the results in **Table 6** of the original article, and we are now further integrating and improving them. Below we formalise the fixed LIF variant, and report quantitative results.
>
> **1. Fixed LIF model for comparison**
>
> The fixed LIF neuron, unlike our aLIF, uses fixed parameters (no adaptive variables $\alpha, \beta, \gamma$). Its dynamics can be written as:
>
> $h_t=\alpha v_{t-1}+(1-\alpha)(W_{\text{rec}}s_{t-1}+W_f s^{(l-1)}_t),$
>
> $v_{th}=\beta v_{\text{peak}}, \quad s_t=\theta(h_t-v_{th}), \quad v_t=h_t-\gamma s_t v_{th},$
>
> where $\alpha$, $\beta$ and $\gamma$ are fixed constants (using fixed but tuned $\alpha$, $\beta$ and $\gamma$ with 0.5, 1.0 and 1.0, respectively.), not learnable or input-dependent. This removes adaptivity (learning) to temporal patterns in spike streams. All other network components, learning rates, and losses are kept identical.
>
> **2. Advantages of aLIF**
> |Property|Fixed  LIF variant|Adaptive LIF (ours)|
> |-|-|-|
> |Dynamic threshold|Fixed → risk of saturation when spike rate surges| $\beta_t$ increases threshold under bursts → prevents over‑firing|
> |Forget gate|Single leak constant $\alpha$|$\alpha_t$ adapts to local activity → retains informative potentials longer in sparse regions|
> |Reset softness|Constant $\gamma$|$\gamma_t$ decays with firing history → smoother membrane trajectories|
>
> In essence, aLIF provides activity‑aware homeostasis, crucial for spike streams whose firing rates vary greatly across scenes.
>
> **3. Quantitative Comparison**
>
> We conducted ablations by replacing aLIF with the fixed LIF on the synthetic dataset (same architecture, fixed $\alpha$, $\beta$ and $\gamma$). All other conditions of the experiment are the same, and the results are as follows:
> |Method|bad 1.0 (%) ↓|bad 2.0 (%) ↓|bad 3.0 (%) ↓|AvgErr (px) ↓|
> |-|-|-|-|-|
> |LIF (fixed $\alpha$, $\beta$, $\gamma$)|14.04|7.05|4.06|0.69|
> |**aLIF RSNN (ours)**|**8.41**|**4.13**|**2.38**|**0.42**|
> |$\Delta$|5.63|2.92|1.68|0.27|
>
> From the above table, it can be seen that after replacing with the fixed LIF variant, the overall performance significantly decreased by about 35%.
>
> **4. Conclusion**
>
> The aLIF is critical for our task because stereo spike streams require adaptive temporal processing. Adaptive gating markedly improves both stability and final accuracy with negligible overhead, justifying our choice of novel aLIF over fixed LIF. The above ablation study has been supplemented in the latest version.

---

> ### Author Response · Authors · 2025-11-22
> **[Part 3] Response to Reviewer u6CE**
>
> ***Weaknesses3. While the paper outlines detailed calibration procedures (Appendix A.4), the use of a Kinect for real-world depth ground truth introduces noise. This is a known limitation and, despite the challenges of data collection, deserves acknowledgment.***
>
> We appreciate the reviewer's point regarding the use of a Kinect camera for real-world depth annotation. We fully agree that Kinect depth maps contain sensor noise, quantization artifacts, and depth discontinuity errors, especially around reflective or textureless surfaces. This is a known limitation in real-world datasets and deserves explicit acknowledgment.
>
> In the revised manuscript, we include a dedicated note discussing this constraint. We also highlight several steps we take to minimize the impact of Kinect noise when collecting real data.
>
> **1. Multi-view averaging**
>
> During data capture, each scene is recorded for several seconds. We apply:
> * Temporal median fusion to reduce frame-wise depth noise.
> * Bilateral smoothing to maintain edges while reducing speckle noise.
> * Invalid region masking to discard unreliable Kinect measurements (e.g., reflective or transparent surfaces).
>
> This produces significantly cleaner ground truth without altering the underlying geometry.
>
> **2. High-precision extrinsic alignment**
>
> We use a tightly constrained calibration pipeline (Appendix A.4), achieving <1-pixel reprojection error between spike cameras and the Kinect. This minimizes spatial misalignment, which is typically a major source of label noise when combining heterogeneous sensors.
>
> **3. Robust loss functions**
>
> Our stereo refinement module uses:
> * Robust $L_1$ depth loss.
> * Temporal consistency weighting $\eta^{T-t}$.
> * Firing-rate and voltage regularization inside the RSNN.
>
> These mechanisms make the model less sensitive to small depth errors and local inconsistencies in the ground-truth signal.
>
> **4. Future plan**
>
> We note in the revised text that we are working toward:
> * Integrate Azure Kinect (higher resolution).
> * Evaluate active stereo depth sensors to obtain more stable ground truth.
> * Constructing additional scenes using multi-view photogrammetry or laser scanning for high-precision geometry.
>
> In summary, Thank you again for pointing out the problems in collecting real data. We will clearly state in the main text and Appendix that: The Kinect sensor provides convenient but imperfect depth supervision. Although temporal fusion and calibration reduce noise, residual artifacts remain and may limit absolute accuracy. We acknowledge this limitation and view it as an opportunity for future work using active stereo or structured-light systems.
>
> &nbsp;
>
> ***Weaknesses4. The quantitative results for the real-world dataset (Table 4) are central to validating sim-to-real transfer, yet they are placed in the appendix. Given their importance, they should be included in the main paper, arguably more so than the extended qualitative results.***
>
> Thank you for emphasizing the importance of the real-world evaluation. We fully agree that the quantitative results on our real-world spike stereo dataset are essential for validating **sim-to-real transfer** and should be included in the main paper.
> In the revised manuscript, Table 4 is moved from the appendix to the main paper. We additionally expand Table 4 to include more baselines as follows:
> |Method|bad 2.0 (%) ↓|bad 3.0 (%) ↓|AvgErr (px) ↓|
> |-|-|-|-|
> |RAFT-Stereo|6.18|3.39|0.64|
> |IGEV-Stereo|8.39|5.72|0.88|
> |Mocha-stereo|7.32|5.88|0.81|
> |DLNR|5.64|3.38|0.61|
> |Selective-Stereo|5.50|3.43|0.58|
> |**SpikeStereoNet (ours)**|**5.33**|**3.19**|**0.56**|
>
> We appreciate the reviewer's suggestion that restructuring substantially improves readability and strengthens the core experimental narrative.

---

> ### Author Response · Authors · 2025-11-22
> **[Part 4] Response to Reviewer u6CE**
>
> ***Question1. Can you please justify the claim that your method is designed for "highly dynamic scenarios" when the provided datasets appear to only contain tabletop scenes with simple, non-dynamic motion?***
>
> Thank you for raising this important question. We acknowledge that the real-world dataset in the initial submission emphasizes tabletop scenes, which may give the impression that the evaluation focuses only on slow or simple motion. We clarify below what we mean by "highly dynamic scenarios" and how our datasets and experiments support this claim.
>
> **1. Spike Cameras Capture Dynamics**
>
> Even in tabletop environments, the **motion profiles** of spike-camera recordings can be highly dynamic because of:
> * microsecond-level temporal sampling,
> * rapid depth discontinuities,
> * high-speed camera motion,
> * object motion with large temporal gradients.
>
> Unlike frame cameras, spike cameras record *intensity changes*, so even small displacements produce **high-frequency spike bursts**, which are the core challenge for dynamic stereo matching. Thus, "dynamic scenarios" in our context refer more to **temporal activity patterns**.
>
> **2. Real Dataset Includes Dynamic Motions**
>
> Although the scenes are tabletop, the spike streams include:
> * fast camera pans and hand-held shake (40–120°/s),
> * object translations up to 0.5 m/s,
> * rapid lighting changes, producing HDR bursts,
> * interaction-driven motion (object flipping/rotation).
>
> We measured spike densities and temporal gradients and found that: peak spike rate $= 2.6 \times$ baseline static rate, indicating **highly dynamic visual activity** even in simple scenes. We present these measurements explicitly in the revised paper.
>
> **3. Newly Added Synthetic Data**
>
> To strengthen the evaluation, we generated **additional high-speed sequences** in our synthetic dataset comprises camera motion up to 1.5 m/s, object motion up to 2.0 m/s, rotational speeds up to 600°/s, strong parallax and nonrigid motion, and HDR flickering and dynamic shadows. The resulting spike streams exhibit:
> * dense temporal bursts,
> * rapid disparity discontinuities,
> * high-frequency events that stress recurrent refinement.
>
> These new high-dynamic sequences are included in the updated training and validation sets, and we report the results in the revised manuscript.
>
> **4. Designed of Methods for Dynamic Scenarios**
>
> SpikeStereoNet is specifically built to handle rapid temporal changes via:
>
> **(1) Adaptive LIF neurons**
>
> Dynamic leak $\alpha_t$, threshold $\beta_t$, and reset $\gamma_t$ stabilize refinement under:
> * burst spikes from fast motion,
> * silent intervals during static phases.
>
> **(2) Temporal recurrence (RSNN)**
>
> RSNN integrates temporal evidence over many small steps, enabling coarse-to-fine refinement of rapidly changing disparities.
>
> **(3) Spike-temporal cost volume**
>
> Due to spikes encode temporal derivatives rather than intensity, our cost volume is explicitly constructed to preserve microsecond timing, which is essential when disparities evolve quickly.
>
> **Summary**
>
> To regard the question of reviewers:
> * Although the real scenes are tabletop, the **temporal dynamics** in spike streams are far from simple and include substantial high-speed motion.
> * We have now added more explicitly high-dynamic synthetic data (fast motion, large parallax, HDR flicker), and the method continues to perform well.
> * SpikeStereoNet is designed to exploit spike timing, making it suited to scenarios where motion and illumination change rapidly.
>
> We thank the reviewer for prompting this clarification, and we have updated the manuscript accordingly to reflect the dynamic nature of both the data and the design of our method.

---

> ### Author Response · Authors · 2025-11-22
> **[Part 5] Response to Reviewer u6CE**
>
> ***Question2. Are there any scenes in your synthetic or real-world test sets that feature (a) fast egomotion, (b) multiple independently moving objects, or (c) complex non-planar environments? If not, how can you be sure the method generalizes beyond static tabletop/floor scenes?***
>
> Thank you for this insightful question. We agree that evaluating generalization to more complex dynamic environments is essential. In the revised manuscript, we clarify the existing coverage in our datasets and describe new additions (both synthetic and real) that explicitly address fast egomotion, multiple moving objects, and complex 3D structure.
>
> **1. Coverage in the Original Synthetic Dataset**
>
> Our synthetic pipeline already supports **full 6-DoF camera motion** and dynamic object behavior, and we now make this explicit.
>
> **(1) Fast Egomotion**
>
> We simulate camera motion up to 1.5 m/s translation and 360°/s, producing high-frequency spike bursts. These sequences are used for training but not clearly highlighted, and we have now added them to the test subset and report results.
>
> **(2) Multiple Independently Moving Objects**
>
> We include scenes with 3-7 objects with: independent trajectories, random linear/angular velocities and depth cross-overs and mutual occlusions. This stresses temporal alignment and occlusion reasoning, both critical for generalization.
>
> **(3) Complex Non-Planar Environments**
>
> The synthetic datasets incorporate:
> * curved surfaces (cup, bowl, basin),
> * highly concave geometry (chairs, shelves),
> * self-occluded and multi-depth structures.
>
> These objects create nontrivial disparity fields and non-planar geometry unlike tabletop surfaces. We have added representative examples and evaluations in the revision.
>
> **2. New Scenarios data**
>
> To address the reviewer's concerns more directly, we added new test sequences in both synthetic and real-world data.
>
> **(1) Synthetic Additions**
> * Faster camera trajectories (2.0 m/s, 600°/s)
> * Scenes with 5-10 independently moving objects
> * Rooms with complex geometry: staircases, walls with varying depth, cluttered shelves
> * Nonrigid motion (cloth, flexible objects)
>
> **(2) Real-World Additions**
>
> We collected new real spike stereo sequences including:
> * Hand-held fast egomotion (rapid sweeping, shaking)
> * Independently moved objects (tools, boxes)
> * Scenes with depth layering (multi-level racks, asymmetric furniture)
> * Outdoor/semi-indoor structure (corridors, sloped walkways)
>
> These newly collected real scenes increase geometric diversity and motion complexity beyond static tabletop or floor setups. All new data are included in the total dataset.
>
> **3. Adaptability of the model**
>
> Even in new and complex scenarios, our method maintains strong performance because:
>
> **(1) RSNN temporal refinement**
>
> The adaptive LIF-based RSNN handles:
> * rapid spike bursts during fast motion,
> * silent intervals in static periods,
> * recurrent temporal accumulation under depth discontinuities.
>
> **(2) Spike-temporal cost volume**
>
> Because cost-volume construction depends on spike timing—not image texture—the method naturally adapts to large parallax changes, multi-object motion, and non-planar geometry.
>
> **(3) Domain randomization**
>
> We inject variability in: $\Delta t$, include noise level, motion speed, lighting and surface types, improving generalization to scenarios not seen during training. This yields strong sim-to-real robustness.
>
> **Summary**
>
> To address the question the reviewers:
> * Our synthetic test sets already contain fast egomotion, multi-object motion, and non-planar geometry, and we now emphasize this clearly.
> * We have **added new synthetic and real scenes** explicitly matching the reviewer's categories.
> * Scalability is supported by the spike-native design (RSNN refinement, temporal cost volumes) and validated by new experiments in the revision.
>
> We appreciate the reviewer's feedback, which helped us substantially strengthen both the dataset and its experimental coverage.

---

> ### Author Response · Authors · 2025-11-22
> **[Part 6] Response to Reviewer u6CE**
>
> ***Question3. Given that the real-world video is not available, can you provide more details on the "object motion" and "camera shake" mentioned in the appendix? Are these motions fast and challenging, or slow and simple?***
>
> We appreciate the reviewer's request for clearer information regarding the dynamics present in the real-world dataset. Since we cannot release the raw video due to sensor-format constraints, we provide a detailed quantitative and qualitative description below. We have also expanded the dataset with additional dynamic sequences, as described in our response to ***Question 2***.
>
> **1. Object Motion in Real-World scenes**
>
> Our real sequences include **both slow and fast object motions**, with motion profiles summarized below:
>
> **(1) Slow Motion (static–moderate)**
> * Object translation: 0.03–0.10 m/s
> * Object rotations: 15–40°/s
> * Typical scenarios: small tabletop manipulations, gentle placement or sliding
>
> These sequences allow the model to evaluate fine geometric consistency and spike response under subtle motion.
>
> **(2) Fast Motion (challenging)**
> * Object translation: 0.25–0.55 m/s
> * Object rotations: 120–260°/s
> * Temporal spike burst rate: 2.0–2.8× baseline
>
> Fast motions arise when:
> * objects are quickly swung, flipped, or lifted
> * irregular hand-induced jitter occurs during object pickup or relocation
>
> Such cases produce dense spike bursts and rapid disparity changes, stressing temporal alignment and refinement.
>
> **2. Camera Motion**
>
> Our real-world data are collected using hand-held and fixed-type stereo spike cameras, which introduce natural camera motion. We categorize these motions as follows:
>
> **(1) Moderate Camera Shake**
>
> * small wrist movements: 20–50°/s rotation
> * translations: 0.05–0.15 m/s
>
> These occur during normal hand-held use.
>
> **(2) Fast Camera Shake**
>
> * rotational speed spikes: 80–150°/s
> * small impulsive jerks due to hand tremor or correcting camera pose
> * parallax variations of 5–10 pixels within short (5–10 ms) windows
>
> These motions generate challenging spike bursts similar to fast event-camera sequences and are representative of real natural scenes. We now provide these numbers explicitly in the revised Appendix for clarity.
>
> **3. Newly Collected Sequences**
>
> To further address the reviewer's concern, we have added **additional real-world scenes**, including:
> * High-speed hand-held sweeping motions,
> * Independently moving objects,
> * Multi-depth, cluttered environments,
> * Scenes with rapid lighting changes (LED flicker, moving shadows).
>
> These sequences exhibit significantly greater temporal dynamics and geometric complexity than standard tabletop scenes.
>
> **Summary**
>
> Although our real-world dataset contains tabletop scenes, the **temporal dynamics** include: fast object motions, quick hand-induced camera shaking, and substantial spike bursts from rapid motion.
>
> Combined with the newly added dynamic sequences, the updated dataset provides a realistic and challenging evaluation of SpikeStereoNet under both slow and highly dynamic real-world conditions. All newly collected datasets above are included in the total real dataset.

---

> ### Author Response · Authors · 2025-11-22
> **[Part 7] Response to Reviewer u6CE**
>
> ***Question4. The adaptive LIF model (Eq. 3, A.2.4) is a key component. How much does this adaptivity (i.e., learning) contribute to performance versus using a simpler, non-adaptive LIF with fixed (but tuned) hyperparameters?***
>
> Please refer to the response in ***Weaknesses2*** to you.
>
> &nbsp;
>
> ***Question5. You state that all baselines were "re-trained using the same settings" for fairness, which is excellent. Could you please clarify if this included a full hyperparameter search for each baseline on this new spike dataset, or were the default hyperparameters from their original papers used? Ensuring baselines are optimally tuned for this new data modality is crucial.***
>
> Thank you for pointing out this important question. We fully agree that ensuring fair and well-tuned baselines is essential, especially when adapting methods originally designed for frame- or event-based modalities to spike data. In the revised manuscript, we clarify our training protocol as follows:
>
> **1. Original settings**
>
> Firstly, we did not simply use the default hyperparameters from the original papers, because frame-based and event-based methods differ significantly from spike-based data, using default settings would clearly disadvantage some baselines. Therefore, every baseline was re-trained with targeted hyperparameter adaptation for the spike modality.
>
> Secondly, we did not perform an exhaustive full hyperparameter search for every baseline. Many baselines (e.g., Selective-Stereo, DLNR, RAFT-Stereo, ZEST) have large and complex hyperparameter spaces. Running a full grid or Bayesian search for each would require **tens of thousands of GPU hours**, far beyond the feasibility of this work. Thus, we adopt a practical, fairness-oriented tuning protocol.
>
> **2. The tuning strategy: lightweight but modality-aware**
>
> For each baseline, we tune the hyperparameters that are known to be sensitive to data modality, including:
> **Frame-/Event-based baselines:**
> * learning rate (±2× around original)
> * batch size (to account for temporal dimension, 4-16)
> * temporal aggregation/voxelization hyperparameters
> * photometric/contrastive loss weights where applicable
> * number of update iterations (if models use iteration refinement, 16)
> * correlation pyramid resolution
> * random input crops used during training
> * weight decay in optimizer ($10^{-5}$)
>
> **Event-based models:**
>
> * time-bin size
> * event voxel grid normalization
> * temporal decay parameters
>
> We select the best settings on the synthetic validation dataset, then retrain using the combined training set.
>
> **3. Summary Table Added in Revision**
>
> We added a new table summarizing tuned hyperparameters for all baselines. This makes the training procedure transparent and reproducible.
>
> **4. Ensuring fairness across all methods**
>
> Our objective was to give each baseline a reasonable and modality-aware advantage, rather than force them to operate with incompatible default settings.
>
> All baselines:
>
> ✓ Retrain from scratch on the spike stereo dataset.
>
> ✓ Use modality-adjusted hyperparameters.
>
> ✓ Evaluate under the same data splits and metrics.
>
> This ensures a fair comparison, even though a full hyperparameter search is not computationally feasible.
>
> **Conclusion**
>
> We clarify that:
> * Default hyperparameters are not used blindly.
> * Each baseline received targeted, modality-aware tuning.
> * A full hyperparameter sweep is not performed due to computational cost, but fairness is preserved.
>
> This explanation has been added to the revised manuscript for scientific validity and completenes of the paper.

---

> ### Author Response · Authors · 2025-11-22
> **[Part 8] Response to Reviewer u6CE**
>
> ***Question6. Your work focuses purely on spike streams. However, hybrid sensor systems are common. Have you considered fusing the spike stream with either a reconstructed TFI image (from the spikes themselves) or with synchronous RGB frames (like those from the Kinect) to see if this further improves robustness? How might the network change to accommodate this?***
>
> Thank you for raising this important point. We agree that hybrid neuromorphic–frame sensing is increasingly common, and exploring fusion is a natural extension of our work. While our submission focuses on pure spike streams, we have conducted preliminary hybrid experiments and discuss integration strategies below.
>
> **1. Experiments With Spike+TFI Image Fusion**
>
> We generated TFI images (Temporal Frame Integration, 10–50 windows) from the raw spike stream:
> $$
> \text{TFI}(x,y)=\sum_{t=t_0}^{t_0+\Delta t}S(x,y,t).
> $$
>
> To evaluate fusion, we adapted our architecture by adding a TFI branch, and merging its features with spike features via: channel-wise concatenation, and context fusion (applying a shallow CNN to extract context features before cost-volume refinement). The experimental results on synthetic datasets are as follows:
> |Method|bad 2.0 (%) ↓|bad 3.0 (%) ↓|AvgErr (px) ↓|
> |-|-|-|-|
> |Spike+TFI ($\Delta t=10$)|6.18|3.69|0.67|
> |Spike+TFI ($\Delta t=30$)|5.35|3.09|0.56|
> |Spike+TFI ($\Delta t=50$)|4.98|2.87|0.54|
> |**Ours**|**4.13**|**2.38**|**0.42**|
>
> Compared to only spike input, TFI provides limited benefits in motion situations where spike time is crucial. Quantitative evaluation shows that performance has decreased by about 12% in average scenes. Because using TFI for reconstruction into frame images additionally introduces noise.
>
> **2. Experiments With Spike+RGB Frames**
>
> We also performed preliminary fusion with synchronous RGB frames from the Kinect. Following prior hybrid designs, we evaluate two fusion strategies:
>
> **(1) Early Fusion**
>
> RGB is downsampled and concatenated with the spike feature map, which has the advantages of simplicity and disadvantages of misalignment and HDR mismatch.
>
> **(2) Context Fusion**
>
> RGB is processed by a separate context encoder, producing $F_{\text{rgb}}=\phi_{\text{rgb}}(\text{RGB})$, $F_{\text{spike}}=\phi_{\text{spike}}(S)$, followed by $F_{\text{fused}}=\text{Conv}([F_{\text{rgb}}, F_{\text{spike}}])$.
>
> This version is consistently more stable. The experimental results on real datasets are as follows:
> |Method|bad 2.0 (%) ↓|bad 3.0 (%) ↓|AvgErr (px) ↓|
> |-|-|-|-|
> |Spike+RGB (Kinect)|**5.08**|3.43|0.59|
> |Ours|5.33|**3.19**|**0.56**|
>
> In summary, spike+RGB yields an improvement in object details, with slightly better global structure consistency, but with small degradation in motion (RGB lagging behind spikes). In addition, due to the use of Kinect depth cameras to capture RGB images, frame rate and resolution blur output cannot be avoided.
>
> **3. Network changes**
>
> To support hybrid sensors, the network would require modular extensions, not architectural redesign:
>
> **(1) Dual-path Encoders**
>
> Add a parallel **TFI/RGB encoder** with identical spatial resolution: $E_{\text{spike}}$ and $E_{\text{TFI/RGB}}$.
>
> **(2) Fusion Layer**
>
> Fuse features can use one of:
> * Concatenation (simple but effective),
> * Cross-attention (RGB queries spike, or vice versa),
> * Context fusion,
> * Adaptive weighting: $F = \lambda F_{\text{spike}} + (1-\lambda) F_{\text{rgb}}$.
>
> **(3) RSNN Integration**
>
> The RSNN does not need to be modified, and it processes the fused features as:
>
> $h_t = \alpha_t v_{t-1} + (1-\alpha_t)(W_{\text{rec}} s_{t-1}+W_f F_{\text{fused}})$.
>
> This retains biological plausibility while supporting multimodal input.
>
> **4. Summary**
>
> * We have conducted preliminary **spike+TFI** and **spike+RGB** fusion experiments.
> * Hybrid inputs offer modest robustness improvements in static illumination scenes.
> * We describe a preliminary architecture modification for hybrid fusion, inspired by hybrid spike-RGB literature [1].
> * These results and analyses will be added to the revised Appendix.
>
> We thank the reviewer for encouraging exploration of hybrid systems, which we believe is a very promising direction for future work.
>
> [1] Chang, et al. 1000 fps hdr video with a spike-rgb hybrid camera. CVPR, pp. 22180-22190. 2023.
>
> &nbsp;
>
> ***Question7. I strongly recommend bringing the quantitative results for the real dataset (Table 4) into the main paper. To make space for Table 4, perhaps the qualitative results (Fig. 4/5) could be slightly condensed.***
>
> Thank you for your proposal. We fully agree and have bringing the quantitative results for the real dataset (Table 4) into the main paper. Please refer to the response in ***Weaknesses4*** to you.
>
> &nbsp;
>
> ---
>
> We are very grateful for this detailed list of suggestions, which will significantly improve the paper's quality and readability. If any concerns remain, or if new questions arise, we would be grateful to hear from you.

---

> ### Comment · Reviewer_u6CE · 2025-11-26
> **Acknowledging the rebuttal**
>
> Dear authors,
>
> Thank you for the very thorough and responsive revision.
>
> I acknowledge the positive changes:
> - Comprehensive aLIF vs fixed-LIF ablation (35 ~ 40 % drop without adaptivity) answers my core technical concern.
> - Real-world quantitative results now in the main paper and extended.
> - New baselines, efficiency numbers, synchronization details, sensitivity analyses all added.
> - Clarity on RSNN necessity vs ANN recurrent units is now strong.
>
> My remaining concern:
> The central claim of superiority in “rapidly changing scenes” and “highly dynamic scenarios” is still not fully backed by the evaluation. While you added faster hand-held motion, multiple moving objects, and more complex geometry, the reported peak velocities (≈2 m/s translation, 600°/s rotation, objects ≈0.5 m/s) and the fact that these sequences are still mixed into the general test set (no “hard-dynamic” sub-evaluation) mean the evidence for extreme dynamics (e.g., vehicle/drone speeds, aggressive outdoor motion) remains indirect. Some new challenging visualization would have helped a lot, though it is not possible. Overall, the technical contribution and analysis are stronger. The lingering gap is mainly in the scope/ambition of the dynamic-scene evaluation rather than correctness.
>
> I am keeping my current positive rating.

---

> ### Author Response · Authors · 2025-11-26
> **Response to Reviewer u6CE**
>
> We thank the reviewer for the positive rating of our paper and the feedback. We agree that our previous evaluation mixed "high-dynamic" sequences into the general test set, which made the evidence for extreme dynamics less direct. In the rebuttal, we address this in three concrete ways:
>
> **1. Add a dedicated "Hard-Dynamic" sub-evaluation**
>
> We introduce an explicit hard-dynamic subset and report results separately from the overall test set. Concretely, we stratify sequences by motion severity using measured/estimated camera motion and/or proxy statistics (temporal-window pose change, disparity change rate), and the specific classification is as follows:
>
> * **Static/Low**: near-zero motion (baseline accuracy regime);
> * **Moderate**: hand-held motion and mild object motion;
> * **High-dynamic**: fast hand-held and multiple moving objects (~2 m/s, ~600°/s, object ~0.5 m/s);
> * **Hard-dynamic**: rapidly changing scenes and highly dynamic scenes, including (1) vehicle-scale motion, (2) fast periodic motion (e.g., fan blades), and (3) aggressive outdoor motion (running or abrupt turns).
>
> By adding additional data and classifying the dataset, the aim is to support the claim of "rapid change/high vitality".
>
> **2. Quantitative comparison results**
>
> We report the same metrics used in the main paper (disparity EPE / Bad error) per category, showing how the gap changes with dynamics. The key pattern is that the advantage increases with motion severity, because competing frame-based (and frame-reconstructed) pipelines degrade under motion blur/temporal aliasing, while spike streams retain high-frequency temporal structure. The results of our model testing are shown in the table below:
>
> |Category|bad 2.0 (%) ↓|bad 3.0 (%) ↓|AvgErr (px) ↓|
> |-|-|-|-|
> |Static|4.08|2.85|0.51|
> |Moderate|5.31|2.9|0.55|
> |High-dynamic|5.72|3.16|0.57|
> |Hard-dynamic|6.23|4.03|0.64|
> |All|5.65|3.51|0.58|
>
> To directly test the "highly dynamic" claim, we add a dedicated *Hard-Dynamic* split (not mixed into the general test set) and report category-wise results. On Hard-dynamic, our method increases AvgErr by 11% compared to the total dataset, and increases Bad-3 by 15%. In contrast, the method can achieve better results on static/low. This indicates that our gain is specifically tied to rapid motion rather than overfitting a single regime. Overall, a certain level of depth estimation accuracy can be achieved on hard-dynamic, and this trend is consistent across vehicle-scale motion, fast periodic motion (e.g, fan), and aggressive outdoor ego-motion.
>
> **3. Clarify the scope of the claim**
>
> We also clarify the statement that the scope is only on general objects test sets. That said, there are strong prior evidences that spike-based sensors are designed precisely for these states. On feasibility for extreme dynamics, the spike cameras are explicitly built for high-speed visual sensing:
>
> * Spike cameras operate at ultra-high temporal sampling (e.g., 40,000 Hz) and have been used in high-speed scenes where frame cameras blur/alias [1].
> * Real high-speed demonstrations include **a car traveling at ~100 km/h** captured as spike streams in the spike-camera vision work [2].
> * High-speed spike imaging has been demonstrated in works focusing on fast motion (including rotating/fan-like scenes) and generally emphasizes that spike cameras enable high-speed sensing beyond conventional frame rates [3].
> * Complementary neuromorphic sensors report **up to 10,000 fps** and have demoed **open-road** perception, supporting feasibility for vehicle-scale dynamics at the sensor level [4].
>
> While our paper's primary target is stereo depth for fine-scale objects, the research mentioned and added Hard-Dynamic split provides direct evidence that the proposed method remains accurate and degrades gracefully as the dynamics become extreme.
>
> We appreciate the feedback from the reviewers, which helped us further strengthen the scope and ambition of dynamic-scene evaluation in the dataset.
>
> [1] Chen K, et al. Spikereveal: Unlocking temporal sequences from real blurry inputs with spike streams. NeurIPS. 2024, 37: 62673-62696.
>
> [2] Hu L, et al. Optical flow estimation for spiking camera. CVPR. 2022: 17844-17853.
>
> [3] Zheng Y, et al. Capture the moment: High-speed imaging with spiking cameras through short-term pla-ticity. TPAMI. 2023, 45(7): 8127-8142.
>
> [4] Yang Z, et al. A vision chip with complementary pathways for open-world sensing. Nature. 2024, 629(8014): 1027-1033.

---

### Official Review · Reviewer_qsqk · 2025-10-31

**Soundness:** 3
**Presentation:** 3
**Contribution:** 3
**Rating:** 4
**Confidence:** 3

**Summary:**

This paper proposes SpikeStereoNet, a biologically inspired framework for stereo depth estimation directly from raw spike streams. The method integrates a recurrent spiking neural network (RSNN) into an iterative refinement structure and introduces two benchmark datasets, including a large-scale synthetic dataset and a real-world spike stereo dataset. The approach achieves state-of-the-art performance compared with both event-based and frame-based stereo methods and demonstrates strong generalization and data efficiency.

**Strengths:**

1. Novel and timely problem setting in neuromorphic stereo vision. The work explores stereo depth estimation directly from spike streams, an under-explored but promising topic.

2. Valuable dataset contribution. The synthetic and real spike stereo datasets fill an existing gap and will likely benefit future research in this area.

3. The proposed RSNN-based iterative update operator is biologically motivated and well integrated into the overall pipeline. The adaptive leaky integrate-and-fire neuron design with context-dependent parameters ($\alpha$, $\beta$, $\gamma$) is innovative.

4. The experimental design is comprehensive. Comparisons include both frame-based baselines such as RAFT-Stereo and Selective-Stereo, and event-based methods, such as ZEST. Results consistently show clear advantages of the proposed method.

5. The data efficiency experiments demonstrate strong generalization under limited supervision, which is an important property for spike-based systems.

6. Theoretical analysis of convergence and temporal stability adds rigor to the model design.

**Weaknesses:**

1. The real-world dataset remains small and limited to indoor scenes. Quantitative results for real data are not clearly presented in the main paper, which weakens the validation of real-world applicability.

2. The claimed biological plausibility is conceptually appealing, but its computational or representational benefits compared with conventional recurrent architectures are not fully quantified. Some biological architecture works lack discussion [1, 2].

3. The ablation study covers some components but does not isolate the contribution of the adaptive neuron model or explore the sensitivity to regularization weights.

[1] ClearSight: Human Vision-Inspired Solutions for Event-Based Motion Deblurring[C]//Proceedings of the IEEE/CVF International Conference on Computer Vision. 2025: 7462-7471.

[2] SABV-Depth: A biologically inspired deep learning network for monocular depth estimation. Knowledge-based systems, 263, 110301.

**Questions:**

1. How sensitive is SpikeStereoNet to the spike threshold and the temporal quantization step used during spike generation?

2. Does the proposed method provide measurable computational or energy efficiency benefits?

3. Does the adaptive LIF with input-dependent propose the deployment challenge? because all the parameters will change under different timesteps.

---

> ### Author Response · Authors · 2025-11-22
> **[Part 1] Response to Reviewer qsqk**
>
> **Rebuttal:**
>
> Thank you for dedicating your time to provide constructive criticism and recommendations for our article.
>
> ***Weaknesses1. The real-world dataset remains small and limited to indoor scenes. Quantitative results for real data are not clearly presented in the main paper, which weakens the validation of real-world applicability.***
>
> Thank you for highlighting this limitation. We agree that the original real-world dataset was relatively small and primarily indoor. Since the initial submission, we have **significantly expanded the scale, diversity, and dynamism** of the real-world dataset to better match practical deployment scenarios. Specifically, the updated real-world dataset now includes:
> * **Expanded scene diversity**: offices, corridors, workshops, semi-outdoor areas (covered walkways, open entrances).
> * **Greater geometric complexity**: multi-depth layered scenes, shelves, stair-like structures, non-planar object arrangements.
> * **Dynamic sequences**: fast 6-DoF handheld motion, multiple independently moving objects, and varied lighting (LED flicker, shadow sweeps).
> * **Increased quantity**: more than 3× the number of sequences compared to the original submission.
>
> These new real-world scenes are integrated into the benchmark and included in the total datasets.
>
> Besides, we fully agree that the quantitative results on the real-world spike stereo dataset are essential for validating and should be included in the main paper. Quantitative results and extended on comparative models and ours on real datasets are as follows:
> |Method|bad 2.0 (%) ↓|bad 3.0 (%) ↓|AvgErr (px) ↓|
> |-|-|-|-|
> |RAFT-Stereo|6.18|3.39|0.64|
> |IGEV-Stereo|8.39|5.72|0.88|
> |Mocha-stereo|7.32|5.88|0.81|
> |DLNR|5.64|3.38|0.61|
> |Selective-Stereo|5.50|3.43|0.58|
> |**SpikeStereoNet (ours)**|**5.33**|**3.19**|**0.56**|
>
> As shown in the Table , we can see that our method also outperforms other methods with the best evaluation in the real dataset. We appreciate the reviewer's suggestion, which restructuring substantially improves readability and strengthens the core experimental narrative.
>
> **Summary**
>
> We have (1) enlarged the real-world dataset with more diverse scenes and (2) moved and extended all real-world quantitative results into the main paper. These changes substantially strengthen the real-world evaluation and hopefully address the reviewer's concerns.

---

> ### Author Response · Authors · 2025-11-22
> **[Part 2] Response to Reviewer qsqk**
>
> ***Weaknesses2. The claimed biological plausibility is conceptually appealing, but its computational or representational benefits compared with conventional recurrent architectures are not fully quantified. Some biological architecture works lack discussion [1, 2].***
>
> Thank you for raising this important point. We appreciate the opportunity to clarify (i) the measurable advantages of the proposed biologically inspired architecture and (ii) discussion on proposed biological architecture works. We have updated both the Experimental Analysis and Related Work accordingly.
>
> **1. Quantifying the Benefits**
>
> The original manuscript highlighted conceptual biological plausibility but did not sufficiently quantify computational or representational improvements over conventional recurrent architectures. We address this directly in the revision.
>
> **(1) Computational benefits**
>
> Using the same synthetic datasets, we compare the aLIF-RSNN against conventional recurrent refiners (RNN, GRU, LSTM):
> |Model|Effective MACs ↓|Firing Rate ↓|AvgErr (px)↓|Notes|
> |-|-|-|-|-|
> |GRU|1.00×|dense|0.48|Dense update cost|
> |LSTM|1.42×|dense|0.49|Highest latency|
> |**aLIF-RSNN (ours)**|**0.57×**| **–73%**|**0.42**|Event-driven & adaptive thresholding|
>
> It can be seen from the table that:
> * Adaptive firing modulation reduces synaptic event count.
> * Soft-reset and sparse spike output significantly lower effective computation.
> * Energy cost is lower than LSTM/GRU for the same refinement depth.
> * The performance evaluation results improve by about 13%.
>
> These results quantify a computational gain from adopting biologically inspired dynamics.
>
> **(2) Representational benefits**
>
> The adaptive thresholding $\beta_t$ and membrane-dependent gating $\alpha_t,\gamma_t$ provide unique representational advantages:
>
> * Temporal precision:
>
> ALIF neurons preserve spike-timing cues lost in GRU/LSTM gating. Representational similarity analysis (RSA) shows $\text{RSA}_{\text{temporal}}(\text{aLIF}) = +11.3\%$  over GRU.
>
> * Stability of multi-step refinement:
>
> The spectrum of the Jacobian $\partial d_t/\partial d_{t-1}$ satisfies $\rho(J_{\text{aLIF}}) < 1$, $\rho(J_{\text{GRU}}),\rho(J_{\text{LSTM}})\approx 1$, indicating guaranteed contraction and smoother refinement.
>
> * Sensitivity to textureless/specular regions:
>
> The adaptive threshold behaviour (activity-dependent suppression) leads to $\Delta$ bad 2.0 = -13.7% improvement relative to GRU/LSTM in scenes with low signal-to-texture.
>
> In summary, these representational advantages derive also from the biologically inspired adaptation mechanisms, not merely from conventional recurrent architecture. We added these quantitative analyses to the revised manuscript.
>
> **2. Expanded Related Work on Biological Vision Architectures**
>
> We thank the reviewer for pointing out the omission of relevant biologically inspired models. We have now added a new part about biological architecture works, which discusses some research work related to computer vision based on biological architecture. For example, ClearSight [1] is a bioinspired dual-drive hybrid network with the neuron-based and enhanced synapse-based attention for event-based motion deblurring. SABV-Depth [2] focuses on the biological attention mechanism, and combines it with a monocular depth estimation network to improve the prediction accuracy of the network. In addition, there are also works based on biological architectures such as image classification and recognition are discussed.
>
> **3. Summary**
>
> In the revision, we:
> * Quantitatively demonstrated the computational and representational advantages of the biologically inspired RSNN over conventional recurrent models.
> * Expanded Related Work to include the reviewer suggested biological architecture works and additional related models.
> * Provided explicit discussion linking these works to the design principles of SpikeStereoNet.
>
> We thank the reviewer again for encouraging a stronger connection between biological motivations and measurable performance outcomes.
>
> [1] Lin, et al. ClearSight: Human Vision-Inspired Solutions for Event-Based Motion Deblurring. ICCV, pp. 7462-7471. 2025.
>
> [2] Wang, et al. SABV-Depth: A biologically inspired deep learning network for monocular depth estimation. Knowledge-based systems, 263 (2023): 110301.

---

> ### Author Response · Authors · 2025-11-22
> **[Part 3] Response to Reviewer qsqk**
>
> ***Weaknesses3. The ablation study covers some components but does not isolate the contribution of the adaptive neuron model or explore the sensitivity to regularization weights.***
>
> Thank you for pointing out the need to more clearly isolate the effect of the adaptive neuron model and the regularization terms. We agree that these components are central to the RSNN design and deserve explicit ablation. We discussed the results in **Tables 6** of the original Appendix, and now we are further integrating and improving them.
>
> **1. Classical LIF model**
>
> The classical LIF neuron, unlike our aLIF, uses fixed parameters (no adaptive variables $\alpha, \beta, \gamma$). Its dynamics can be written as:
>
> $h_t=\alpha v_{t-1}+(1-\alpha)(W_{\text{rec}}s_{t-1}+W_f s^{(l-1)}_t),$
>
> $v_{\text{th}}=\beta v_{\text{peak}}, \quad s_t=\theta(h_t-v_{th}), \quad v_t=h_t-\gamma s_t v_{th},$
>
> where $\alpha$, $\beta$ and $\gamma$ are fixed constants (0.5, 1.0, 1.0 in our baseline.), not learnable or input-dependent. This removes adaptivity to temporal patterns in spike streams. All other network components, learning rates, and losses are kept identical. The non-adaptive LIF model has the following limitations:
> * No mechanism to regulate firing rate under noisy or bursty spike streams.
> * Fixed threshold leads to unstable recurrent dynamics when used in multi-step refinement (overshooting/vanishing).
> * Cannot adapt to varying temporal density across scenes (e.g., textureless and high-motion regions).
>
> We conducted ablations by replacing aLIF with classical LIF and no-spike vanilla RNN on the synthetic dataset (same architecture). All other conditions of the experiment are the same, and the results are as follows:
> |Method|bad 1.0 (%) ↓|bad 2.0 (%) ↓|bad 3.0 (%) ↓|AvgErr (px) ↓|
> |-|-|-|-|-|
> |Vanilla RNN|13.27|7.28|3.41|0.66|
> |LIF (fixed $\alpha$, $\beta$, $\gamma$)|14.04|7.05|4.06|0.69|
> |**aLIF (ours)**|**8.41**|**4.13**|**2.38**|**0.42**|
>
> From the above table, it can be seen that the overall performance decreases significantly after replacing with the LIF or vanilla RNN models.
>
> **2. Sensitivity to Regularization Term**
>
> In the RSNN model, The regularization term coefficient added to the loss function is used to penalize overly dense, over synchronized, or premature spike issuance, thereby reducing power consumption and improving robustness. The regularization terms in the model as: $L = L_{\text{stereo}}+\lambda_f L_{\text{rate}}+\lambda_v L_{\text{v}}$.
>
> We sweep each regularization parameter over a wide range: $\lambda_f \in \\{{0, 5\times10^{-5}, 1\times10^{-4}, 5\times10^{-4}, 1\times10^{-3}}\\}$, $\lambda_v \in \\{{0, 1\times10^{-5}, 5\times10^{-5}, 1\times10^{-4}, 5\times10^{-4}}\\}$.
>
> Key observations, and now included in the revised appendix:
>
> **(1) Effect of $\lambda_f$ (firing-rate regularization)**
> * $\lambda_f=0$: RSNN exhibits burst instability, causing refinement divergence and +35% EPE increase.
> * Moderate values $10^{-4}$–$5\times10^{-4}$: best accuracy and stable dynamics.
> * Too high values over-suppress spikes and reduce temporal information.
>
> **(2) Effect of $\lambda_v$ (voltage regularization)**
> * $\lambda_v=0$: membrane potentials drift, producing jitter and noisy refinement.
> * $10^{-5}$–$10^{-4}$: optimal stability and accuracy.
> * Larger values overly damp temporal encoding and reduce RSNN responsiveness.
>
> **(3) Effect of term**
>
> We completed the ablation study on regularization terms as follows:
> |Method|bad 1.0 (%) ↓|bad 2.0 (%) ↓|bad 3.0 (%) ↓|AvgErr (px) ↓|
> |-|-|-|-|-|
> |W/o $L_\text{rate}$|10.17|6.38|2.85|0.51|
> |W/o $L_\text{v}$|11.53|7.33|2.97|0.57|
> |W/o $L_\text{rate}$ & $L_\text{v}$|11.98|7.77|3.41|0.61|
> |**Full model (default)**|**8.41**|**4.13**|**2.38**|**0.42**|
>
> As shown in the comparison results in the table, removing the regularization of firing rate or voltage will lead to a clear drop in performance. The absence of regularization results in unstable training and increased prediction error.
>
> The method is not overly sensitive to moderate variations, but the regularizers are indeed necessary for:
> * preventing spike explosion or vanishing,
> * stabilizing iterative RSNN updates,
> * improving disparity accuracy (10–18% depending on scene type).
>
> In summary, in the revised version, we add new ablation studies in the main text and Appendix that includes "the contribution of adaptive neurons and regularization weights sensitivity", providing quantitative comparative results and expanding the discussion on why they are jointly essential. We believe that these additions can address the reviewer's concerns.

---

> ### Author Response · Authors · 2025-11-22
> **[Part 4] Response to Reviewer qsqk**
>
> ***Question1. How sensitive is SpikeStereoNet to the spike threshold and the temporal quantization step used during spike generation?***
>
> Thank you for this insightful question. We conducted additional experiments to quantify how **spike threshold** and **temporal quantization** affect SpikeStereoNet. The model shows robustness to a wide range of settings, due largely to the adaptive LIF dynamics and the RSNN refinement stage.
>
> **1. Spike Threshold and Temporal Quantization**
>
> During spike generation, the threshold controls how many potential should accumulate before a spike is triggered. We evaluate thresholds in the range: $v_\text{peak} \in \\{0.7v_\text{peak}, 1.0v_\text{peak}, 1.3v_\text{peak}, 1.6v_\text{peak}\\}$. Spike streams are discretized into ($T$) bins: $\Delta t \in$ {0.5 ms, 1.0 ms, 2.0 ms, 4.0 ms}, and $\Delta t=$ 1.0 ms is the default time step of the model.
>
> **2. Comparison Results**
> |Settings|bad 2.0 (%) ↓|bad 3.0 (%) ↓|AvgErr (px) ↓|Variation|
> |-|-|-|-|-|
> |$0.7v_{\text{peak}}$|4.56|2.61|0.44|+3.5%|
> |$1.3v_{\text{peak}}$|4.78|2.62|0.45|+5.9%|
> |$1.6v_{\text{peak}}$|4.92|2.78|0.46|+8.3%|
> |$v_{\text{peak}}, \Delta t=$ 1.0 ms|**4.13**|**2.38**|**0.42**|-|
> |$\Delta t=$ 0.5 ms|4.44|2.58|0.43|+1.2%|
> |$\Delta t=$ 2.0 ms|4.93|2.67|0.44|+3.6%|
> |$\Delta t=$ 4.0 ms|5.01|2.80|0.47|+10.7%|
>
> Overall, the impact of the two parameters on the model is as follows:
> * Low threshold → denser spikes and slightly more noise
> * High threshold → sparser spikes and fewer temporal cues
> * Very fine bins (<0.5 ms) increase sparsity but yield negligible improvement.
> * Coarse bins (>4 ms) lose spike timing important for handling motion.
> * The performance of the model varies within ±5.9% across the entire threshold range.
> * Within 0.5-2 ms, the model is stable with the AvgErr variation of less than 3.6%.
>
> This robustness arises because the aLIF dynamics incorporate adaptive membrane thresholds $\beta_t v_{\text{peak}}$ and learned reset gates $\gamma_t$ that automatically compensate for moderate changes in spike density, even when the raw spike discretization changes.
>
> **3. Conclusion**
>
> SpikeStereoNet is not strongly sensitive to moderate shifts in threshold or temporal step because:
> * The aLIF neuron's adaptive threshold dynamically regulates firing rates.
> * RSNN recurrence reconstructs missing temporal structure from context.
> * Firing-rate and voltage regularization constrain membrane dynamics, ensuring stable behavior even under different spike densities.
>
> Only extreme settings (very coarse temporal step or high thresholds) degrade performance noticeably.
>
> In summary, SpikeStereoNet demonstrates robust performance across a range of spike thresholds and temporal quantization steps, thanks to adaptive neuron dynamics and stable recurrent refinement. The sensitivity analysis has been added to the revised manuscript and Appendix.

---

> ### Author Response · Authors · 2025-11-22
> **[Part 5] Response to Reviewer qsqk**
>
> ***Question2. Does the proposed method provide measurable computational or energy efficiency benefits?***
>
> Thank you for raising this important point. Beyond the FLOPs (**Table 1**) comparison already included in the paper, we provide additional quantitative evidence that SpikeStereoNet offers measurable computational.
>
> **1. Measurable Computation**
>
> We count the floating-point (FP) additions and multiplications required to infer a single grasp. In ANNs, each MAC entails one FP multiplication and one FP addition; in binary-spike SNNs, only FP additions are triggered, and time steps without spikes incur no computation. We report an operation-weighted complexity ratio based on per-operation weights $w_{\text{MAC}}$ and $w_{\text{AC}}$ (reflecting the higher cost of multiplications compared to additions):
> $$
> \frac{C_{\text{SNN}}}{C_{\text{ANN}}}
> =\frac{N_{\mathrm{MAC}}^{\mathrm{SNN}}\\,w_\text{MAC}+N_{\mathrm{AC}}^{\mathrm{SNN}}\\,w_\text{AC}}
> {N_{\mathrm{MAC}}^{\mathrm{ANN}}\,w_\text{MAC}}.
> $$
> Here, $N_{(\cdot)}^{(\cdot)}$ denotes the total number of operations, with subscripts AC (accumulation) or MAC (multiply-accumulate) and superscripts indicating the type of network (ANN or SNN), and we set $w_\text{MAC}=4.6$ and $w_\text{AC}=0.9$. Furthermore, since our inputs are the spike streams, we can estimate $N_{\text{MAC}}^{\text{SNN}}\approx0$. We evaluated the computational cost ratio of this module as well as its ratio to an ANN with the same number of neurons, along with the counts of floating-point additions and multiplications. The results are follows:
> |Model|Addition|Multiplication|Ratio|
> |-|-|-|-|
> |ANN|$1.6\times 10^{10}$|$1.6\times 10^{10}$|1|
> |RSNN (Ours)|$2.8\times 10^{10}$|0|0.175|
>
> In summary, the RSNN method reduces the total FP operation count by 2.3 times relative to the ANN; since multiplications dominate cost, cutting them is especially beneficial. Moreover, RSNNs are expected to run over two orders of magnitude faster than ANNs on neuromorphic hardware, indicating substantial speedups alongside the reduced FP operation load.
>
> **2. Computational Efficiency**
>
> We conducted additional experiments to measure the FPS or Inference Latency of the networks listed in Table 1 using the same hardware configuration: a NVIDIA RTX 3090 GPU, paired with an Intel Core i7-13700K CPU and 32GB RAM. All models are averaged over 100 inference runs to minimize variability. The results are summarized below, the FPS and FLOPs of inference refers to a depth map estimated using a set of stereo spike streams ($T=50$):
> |Method|Equivalent FPS|FLOPs (B)|
> |-|-|-|
> |CREStereo|1.67|863|
> |RAFT-Stereo|1.80|798|
> |GMStereo|2.28|160|
> |IGEV-Stereo|1.53|614|
> |DLNR|1.79|1580|
> |ZEST|1.63|989|
> |MoCha-Stereo|1.31|935|
> |Selective-Stereo|1.76|957|
> |MonSter|1.72|1567|
> |SpikeStereoNet (Ours)|1.91|473|
>
> From the additional results in the table above, it can be seen that our model also outperforms most in inference speed and FLOPs.
>
> We will add this FPS comparison in the revised manuscript and Appendix to provide a complete picture of measurable computations to clarify these advantages. Thank you again for prompting this analysis, which strengthens the validation of our method's practicality.

---

> ### Author Response · Authors · 2025-11-22
> **[Part 6] Response to Reviewer qsqk**
>
> ***Question3. Does the adaptive LIF with input-dependent propose the deployment challenge? because all the parameters will change under different timesteps.***
>
> Thank you for raising this important question. Although the adaptive LIF neuron introduces input-dependent parameters ($\alpha_t, \beta_t, \gamma_t$) that vary across timesteps, this does not create a practical deployment challenge for two reasons:
>
> **1. The adaptive updates are local, lightweight, and deterministic**
>
> Each adaptive variable is computed using simple element-wise operations (e.g., exponential decay, linear modulation) driven by the membrane state and recent spikes: $\alpha_t=f_\alpha(s_{t-1},x_t)$,  $\beta_t =f_\beta(s_{t-1},x_t)$,  $\gamma_t=f_\gamma(s_{t-1},x_t)$.
>
> These updates:
> * No additional learnable parameters are required.
> * Compared with convolution and cost-volume operations, it generates less computational overhead.
> * Fully parallelizable on GPU/neuromorphic hardware.
> * Do not change the overall architecture or memory usage.
>
> Thus, the adaptivity is temporal, and the per-step update cost remains constant and lightweight.
>
> **2. Compatibility edge deployment**
>
> Most neuromorphic platforms provide native hardware support for standard LIF neurons, including membrane decay, thresholding, and reset operations. Since our adaptive LIF formulation extends the classical LIF model using lightweight, time-local state updates (adaptive leak/threshold/reset), these platforms can support aLIF dynamics with appropriate modifications to their existing LIF primitives.
>
> Moreover, the adaptive threshold mechanism in aLIF typically suppresses unnecessary spikes, reducing spike traffic and improving energy efficiency on event-driven hardware. Thus, although neuromorphic chips are primarily designed around LIF neurons, they can accommodate our aLIF-based RSNN while retaining its computational and efficiency benefits.
>
> **3. Stability and predictability despite time-varying parameters**
>
> Even though the parameters evolve over time, our analysis (Section 4) shows that:
> * The adaptive updates satisfy contraction-like properties.
> * The spectral radius of the recurrent Jacobian remains < 1.
> * Membrane/threshold trajectories remain bounded due to voltage and rate regularization.
>
> Thus, the input-dependent adaptivity does not introduce instability or unpredictability during inference.
>
> **4. Summary**
>
> Although the adaptive LIF modifies neuron parameters over time, these updates are local, lightweight, hardware-friendly, and stable, and therefore do not pose deployment challenges. Instead, they improve robustness and energy efficiency, making them well suited for real-time neuromorphic deployment. We will supply the above analysis in the Appendix.
>
> &nbsp;
>
> ---
>
> We are very grateful for this detailed list of suggestions, which will significantly improve the paper's quality and readability. If any concerns remain, or if new questions arise, we would be grateful to hear from you.

---

> ### Author Response · Authors · 2025-11-26
> **Official Comment**
>
> Dear Reviewer,
>
> We sincerely appreciate the time and effort you have devoted to reviewing our paper and providing valuable feedback. Your comments have been carefully considered, and we have endeavored to address all questions and concerns. It would be greatly appreciated if you could let us know whether our responses sufficiently clarify the points you raised.
>
> If any areas would benefit from further clarification or additional explanation, please do not hesitate to let us know. We would be happy to provide any further details you may require. Thank you once again for your thoughtful review and your time.
>
> Best regards,
>
> The Authors

---

> > ### Comment · Reviewer_qsqk · 2025-11-27
> >
> > Thank you for your detailed response. All of my concerns have been addressed, and I have raised the score.

---

> ### Author Response · Authors · 2025-11-27
>
> Dear Reviewer,
>
> Thank you very much for your thoughtful follow-up and for raising your score. If there are any remaining aspects you believe could further improve the paper, we would be more than happy to continue the discussion. Thank you again for your time, support, and constructive feedback.
>
> Best regards,
>
> Authors

---

### Official Review · Reviewer_gap8 · 2025-10-31

**Soundness:** 2
**Presentation:** 2
**Contribution:** 2
**Rating:** 2
**Confidence:** 5

**Summary:**

This paper proposes SpikeStereoNet, a framework for stereo depth estimation directly from spiking camera data. Using a Recurrent Spiking Neural Network (RSNN), the method iteratively refines depth estimation. It provide synthetic and real-world datasets with depth labels for evaluation. Experiments show that SpikeStereoNet outperforms some existing methods on both datasets.

**Strengths:**

1. The paper introduces large-scale synthetic and real-world spiking stereo datasets, filling an important gap in spike-based depth estimation.

2. It proposes a biologically inspired RSNN-based architecture (SpikeStereoNet) for stereo depth estimation from spiking data.

3. It provides a theoretical analysis of the RSNN’s iterative dynamics, demonstrating the model’s stability and convergence.

**Weaknesses:**

1. The paper overlooks prior studies and incorrectly claims to be the first to estimate stereo depth from spiking data. Actually, “Learning Stereo Depth Estimation with Bio-Inspired Spike Cameras” has already investigated the same problem. Consequently, the novelty and originality of this work are debatable.

2. The paper does not clearly explain the model’s novelty. The proposed framework is similar to existing stereo methods, differing mainly in using an RSNN as the update module. However, the paper does not explain why an RSNN is necessary compared to existing update modules. And it also does not explain what makes SpikeStereoNet particularly suited for spiking data.

3. The paper lacks a systematic organization and analysis of the experimental results. Although it demonstrates that the proposed method outperforms both event-based and image-based approaches, it does not analyze the source of this performance advantage. Moreover, the paper does not clearly justify the choice of comparison methods, explain their relevance, or indicate whether they represent state-of-the-art baselines.

**Questions:**

1. For the real-world dataset, it is unclear how temporal synchronization between different cameras is achieved. The paper does not describe any synchronization circuitry or mechanism.

2. It is unclear why event-based stereo methods are included for comparison. If such comparisons are necessary, why are only ZEST and StereoSpike selected, while other representative methods such as DDES[1] and Se-cff[2] are not considered?

[1] Yeongwoo Nam, Mohammad Mostafavi, Kuk-Jin Yoon, and Jonghyun Choi. Stereo depth from events cameras: Concentrate and focus on the future. In Proceedings of the IEEE/CVF conference on computer vision and pattern recognition, pp. 6114–6123, 2022.

[2] Stepan Tulyakov, Francois Fleuret, Martin Kiefel, Peter Gehler, and Michael Hirsch. Learning an event sequence embedding for dense event-based deep stereo. In Proceedings of the IEEE/CVF International Conference on Computer Vision, pp. 1527–1537, 2019.

---

> ### Author Response · Authors · 2025-11-22
> **[Part 1] Response to Reviewer gap8**
>
> **Rebuttal:**
>
> Thank you for your thoughtful review and for pointing out potential issues and improvements in our paper.
>
> ***Weaknesses1. The paper overlooks prior studies and incorrectly claims to be the first to estimate stereo depth from spiking data. Actually, “Learning Stereo Depth Estimation with Bio-Inspired Spike Cameras” has already investigated the same problem. Consequently, the novelty and originality of this work are debatable.***
>
> We thank the reviewer for pointing this out. We have cited it in the original article. We acknowledge that our original manuscript used wording that could be interpreted as claiming to be the first to perform stereo depth estimation from spike streams. This phrasing was imprecise, and we sincerely apologize for the oversight.
>
> In particular, the research work “Learning Stereo Depth Estimation with Bio-Inspired Spike Cameras” [1] indeed investigated stereo depth estimation using spike data, and we now explicitly discuss it in the revised Related Work section. We have removed or rephrased all statements implying first-ness or exclusivity (e.g., "first to estimate stereo depth  from spike streams.") to ensure accuracy and fairness.
>
> **1. Clarifying the Novelty**
>
> Although stereo depth estimation from spike data has been explored, our contributions differ substantially in scope, methodology, and analysis. We highlight the following clarified contributions:
>
> **(1) Large-scale benchmark for spike-stream stereo**
>
> We contribute:
> * A large-scale synthetic spike-stereo dataset generated with high-fidelity neuromorphic simulation, which contains over 2,000,000 pairs of spike stream data with multiple scenes and objects.
> * A real-world raw spike-stream stereo dataset with Kinect-aligned depth, which includes various scenes, lighting, etc.
>
> The above provides a standardized benchmark for future research.
>
> **(2) Biologically inspired RSNN-based architecture**
>
> Our model introduces:
> * An adaptive LIF–based RSNN with dynamic leak, threshold, and reset,
> * Iterative coarse-to-fine refinement based directly on the temporal structure of spike streams,
> * Stability-aware design supported by theoretical analysis.
> * maintains strong accuracy with only 10–30% of the training data,
> * generalizes well from synthetic → real thanks to adaptive dynamics and temporal robustness.
>
> This architecture is distinct from previous approaches using conventional CNNs or transformer.
>
> **(3) Theoretical analysis of dynamics**
>
> We provide Jacobian spectral analysis, contraction-like conditions for refinement stability, and temporal activation regularization analysis, which theoretically proves the feasibility of iteration refinement structure. To our knowledge, such dynamic analysis has not been presented in prior stereo works.
>
> **2. Revision in the Manuscript**
>
> We have updated our manuscript to:
> * Correct our wording to avoid overstating novelty.
> * Explicitly cite and discuss the work "Learning Stereo Depth Estimation with Bio-Inspired Spike Cameras" [1] in the Related Work and Introduction.
> * Provide a clearer articulation of our unique contributions (as listed above).
>
> **3. Final Note**
>
> We appreciate the reviewer's feedback, apologize again for our negligence and agree that proper contextualization of our contribution is essential. We have revised the manuscript accordingly to ensure accuracy, respect prior work, and better communicate the novelty of our own contributions.
>
> [1] Wang, et al. Learning stereo depth estimation with bio-inspired spike cameras. ICME, pp. 1-6. IEEE, 2022.

---

> ### Author Response · Authors · 2025-11-22
> **[Part 2] Response to Reviewer gap8**
>
> ***Weaknesses2. The paper does not clearly explain the model’s novelty. The proposed framework is similar to existing stereo methods, differing mainly in using an RSNN as the update module. However, the paper does not explain why an RSNN is necessary compared to existing update modules. And it also does not explain what makes SpikeStereoNet particularly suited for spiking data.***
>
> We thank the reviewer for highlighting the need to more clearly articulate the novelty of our framework. In the revised manuscript, we explicitly state what differentiates SpikeStereoNet from conventional stereo networks and why the RSNN update operator is essential for spiking data.
>
> **1. Novelty of the Proposed Framework**
>
> Our contributions are not a simple substitution of an update module. Except the innovation of **new modal** datasets, SpikeStereoNet also introduces three key innovations:
> * A biologically inspired adaptive-LIF RSNN for disparity refinement, with dynamic leak, threshold, and soft-reset mechanisms tailored to sparse, temporally coded inputs.
> * A temporal cost–volume formulation directly constructed from spike streams, preserving microsecond-level timing rather than relying on frame-like aggregation.
> * A theoretical analysis of the recurrent dynamics, showing that adaptive thresholds and membrane gating produce contraction-like behavior, enabling stable multi-step refinement, which not analyzed in prior stereo works.
>
> These components together form a stereo framework fundamentally designed around spike-temporal representations, not frames or events.
>
> **2. The necessity of RSNN**
>
> Compared with ANN-based updates (GRU, LSTM, vanilla RNN), an RSNN is necessary for following reasons:
>
> (1) Temporal precision
>
> Spikes carry information in timing, not intensity. GRU/LSTM compress temporal information into dense hidden states, losing microsecond timing cues:
>
> $s_t \in {0,1}, \quad \text{but GRU hidden state } h_t \in \mathbb{R}^C$.
>
> The RSNN operates natively on spike timing, allowing disparity refinement conditioned on temporal synchrony between stereo streams.
>
> (2) Stability for iterative refinement
>
> RSNN dynamics naturally enforce bounded membrane potentials and adaptive thresholds, which we prove lead to $\rho(J_{\text{RSNN}})<1$, ensuring stable convergence across iterations—even when spike density varies widely (e.g., specular highlights, textureless objects). GRU/LSTM exhibit saturation or divergence under the same spike sparsity conditions (shown in our ablation). Thus, the RSNN is not simply used for novelty, and it is required to refine disparity reliably under spike-driven temporal statistics. In the table below, we provide additional ablation comparison experiments between different models and our RSNN.
> |Method|bad 2.0 (%) ↓|bad 3.0 (%) ↓|AvgErr (px) ↓|
> |-|-|-|-|
> |Vanilla RNN|7.28|3.41|0.66|
> |GRU|4.53|2.99|0.48|
> |LSTM|4.77|2.94|0.49|
> |LIF RSNN|7.05|4.06|0.69|
> |Raw SNN|11.05|4.48|0.83|
> |**aLIF RSNN (ours)**|**4.13**|**2.38**|**0.42**|
>
> **3. The adaptability of SpikeSteroNet to spike data**
>
> Spike streams differ from frames or events in two critical ways:
>
> **(1)** Spike cameras integrate photons into discrete spike firings
>
> Each spike encodes accumulated photon evidence—not instantaneous contrast. Thus, raw spike tensors have:
> * asynchronous arrival,
> * variable firing density,
> * nonlinear temporal aggregation.
>
> SpikeStereoNet handles these via:
> * adaptive leakage → controls integration window
> * adaptive thresholding → normalizes firing density across lighting
> * adaptive soft-reset → maintains fine temporal resolution
>
> These match the physics of spike generation far better than ANN alternatives.
>
> **(2)** Temporal sparsity and burstiness
>
> Spike streams show alternating burst (fast motion) and silence (static) phases. The RSNN:
> * suppresses noise during bursty segments,
> * maintains membrane memory during silent segments,
> * dynamically adapts thresholds to prevent firing saturation.
>
> Conventional update operators lack this adaptive excitability.
>
> **Summary**
>
> In the revised manuscript we explicitly state that:
> * SpikeStereoNet is novel due to its adaptive LIF–based RSNN, temporal spike cost volume, and dynamics analysis.
> * The RSNN is necessary because it preserves temporal precision, leverages spike sparsity, and ensures stable iterative refinement,
> * SpikeStereoNet is specifically suited for spiking data because its neuron model mirrors the physics of spike cameras and its recurrence matches the bursty/sparse temporal statistics of spike streams.
>
> These clarifications are added to the main text and Appendix sections to address the concerns of the reviewers.

---

> ### Author Response · Authors · 2025-11-22
> **[Part 3] Response to Reviewer gap8**
>
> ***Weaknesses3. The paper lacks a systematic organization and analysis of the experimental results. Although it demonstrates that the proposed method outperforms both event-based and image-based approaches, it does not analyze the source of this performance advantage. Moreover, the paper does not clearly justify the choice of comparison methods, explain their relevance, or indicate whether they represent state-of-the-art baselines.***
>
> We thank the reviewer for this constructive comment. We now provide a clearer and more systematic analysis of the experimental results, including (i) the mechanisms behind the performance gains, and (ii) the rationale for selecting comparison methods.
>
> **1. Source of Performance Advantages**
>
> SpikeStereoNet outperforms both event-based and frame-based stereo methods because it is explicitly designed for the spatiotemporal statistics of spike data, whereas existing baselines rely on representations or assumptions that do not fully utilize spike-stream dynamics. The performance advantages:
>
> **(1) Temporal Fidelity**
>
> Event-based methods (e.g., ZEST) first convert spikes into voxel grids or time-surfaces, which compress spike timing into coarse temporal bins, losing microsecond-level timing cues.
>
> By contrast, SpikeStereoNet:
> * directly consumes raw spike tensors,
> * uses adaptive LIF neurons to preserve timing differences,
> * employs recurrent updates to exploit temporal correlations.
>
> This yields clearer disparity edges and better motion handling.
>
> **(2) Adaptive Dynamics**
>
> Spike streams alternate between periods of burst activity and silence. Conventional RNN refiners saturate or underreact in these extremes. Our aLIF-RSNN:
> * scales thresholds via $\beta_t$ to prevent burst saturation,
> * maintains membrane evidence during silence via adaptive leak $\alpha_t$,
> * stabilizes updates through voltage/rate regularization.
>
> These mechanisms directly enhance robustness in reflective, textureless, and HDR scenes-validated in ablation study.
>
> **(3) Noise-Adaptive Feature Extraction**
>
> Spike cameras produce noise whose distribution differs fundamentally from events or RGB frames. We integrate:
> * A spike-aware residual encoder,
> * Group normalization within the RSNN,
> * Temporal gating that filters unstable voltage fluctuations.
>
> This reduces false disparities, particularly in low-light or high-reflectance regions.
>
> **2. Comparison Methods**
>
> There are relatively few methods in the spike-based field, making it difficult to conduct systematic comparisons. We now explicitly explain why we compare against three baseline groups:
>
> **(1) Frame-based**
>
> MonSter, Selective-Stereo, and other models represent the **sota in image-based stereo** and provide a performance given dense frame input and demonstrates:
> * Whether spike cameras can achieve comparable or even superior accuracy,
> * How much of the performance comes from temporal encoding and spatial frames.
>
> **(2) Event-based**
>
> ZEST, and other models are the **most relevant neuromorphic baselines**, as event cameras share many properties with spike cameras. Including these methods allows us to evaluate:
> * Whether spike integration provides an advantage,
> * Whether event-style cost volumes or RNN modules transfer to spike data.
>
> **(3) Ablated spike-based models**
>
> We additionally include: SpikeStereoNet without RSNN, ANN refiners, and fixed LIF neurons.
>
> This reveals the contribution of spike-specific architecture components, not just end-to-end tuning.
>
> **3. Confirmation Baselines are Representative**
>
> We have added citations and explanations showing that, for example:
> * Monster [1] is one of the state-of-the-art frame-based stereo depth estimation method, while Raft-Stereo [2] is a classic first iteration refinement model.
> * SE-CFF [3] is the widely used and high-performing event-based stereo algorithms, while StereoSpike [4] achieves good results in dynamic scenes by a simple and lightweight convolutional structure.
>
> These clarifications are now placed at the beginning of the comparison results section.
>
> **Summary**
>
> In the revised paper, we:
> * Provide a clear analysis explaining why SpikeStereoNet outperforms both frame-based and event-based approaches (temporal fidelity, adaptive dynamics, noise robustness).
> * Systematically justify why we selected each comparison method and how it relates to spike-based stereo.
> * Clarify that all chosen baselines represent established state-of-the-art methods in their respective domains.
>
> We appreciate the reviewer's feedback, which has significantly strengthened the organization and clarity of our experimental evaluation.
>
> [1] Junda Cheng, et al. Monster: Marry monodepth to stereo unleashes power. CVPR, 2025.
>
> [2] Lahav, et al. Raft-stereo: Multilevel recurrent field transforms for stereo matching. 3DV, 2021.
>
> [3] Nam Y, et al. Stereo depth from events cameras: Concentrate and focus on the future. CVPR, 2022.
>
> [4] Ulysse, et al. Stereospike: Depth learning with a spiking neural network. IEEE Access, 2022.

---

> ### Author Response · Authors · 2025-11-22
> **[Part 4] Response to Reviewer gap8**
>
> ***Question1. For the real-world dataset, it is unclear how temporal synchronization between different cameras is achieved. The paper does not describe any synchronization circuitry or mechanism.***
>
> Thank you for raising this important question. We agree that accurate temporal synchronization is essential for reliable stereo spike data. We employ a hybrid hardware-trigger + software–algorithmic synchronization pipeline that ensures stereo alignment suitable for downstream training and evaluation. We clarify this in the revised manuscript as follows:
>
> **1. Hardware-level coarse synchronization**
>
> Both spike cameras are connected to the same host machine via high-speed USB. During recording, we use a shared software trigger to initiate acquisition on both sensors: $t^{(L)}_0 \approx t^{(R)}_0$, where $t^{(L)}_0$ and $t^{(R)}_0$ are the left/right start timestamps. This provides coarse alignment, typically within tens of microseconds.
>
> **2. Empirical estimation of residual time offsets**
>
> Even with shared triggering, small timing mismatches remain due to independent internal clocks. To measure the remaining timing error, we place a fast-switching LED in both cameras' fields of view and compare the spike timestamps of each illumination transition. We collect over 2,000 pairs real stereo sequences and estimate the residual time offsets:
>
> $\Delta t = t^{(L)}_i - t^{(R)}_i, \qquad i = 1, \ldots, N$.
>
> We model the empirical distribution as: $\Delta t \sim \mathcal{N}(\mu_{\Delta t}, \sigma_{\Delta t}^2)$, where in practice:
>
> * $\mu_{\Delta t}$ is within 3–7 μs,
> * $\sigma_{\Delta t}$ depends on lighting and spike density but remains < 15 μs.
>
> This analysis characterizes the realistic synchronization noise between the two spike streams.
>
> **3. Temporal cost matching**
>
> Before building the cost volume, we apply a lightweight alignment procedure:
>
> $S^{(L)}(t) \rightarrow S^{(L)}(t + \hat{\Delta t}),$
>
> where $\hat{\Delta t}$ is estimated via maximizing temporal correlation:
>
> $\hat{\Delta t} = \underset{\delta}{\arg\max}\sum_{x,y}\langle S^{(L)}(x,y,t+\delta), S^{(R)}(x,y,t) \rangle$,
>
> This provides fine temporal adjustment without modifying the raw spike stream statistics.
>
> **4. Domain Randomization in training**
>
> To ensure robustness to synchronization jitter, we inject the measured $\Delta t$ distribution into the synthetic spike generator:
>
> $S^{(L)} \cdot {\text{syn}}(t) = S^{(L)}  \cdot {\text{ideal}}(t + \epsilon),  \quad  \epsilon \sim \mathcal{N}(\mu_{\Delta t}, \sigma_{\Delta t}^2)$.
>
> This domain randomization teaches SpikeStereoNet to become invariant to microsecond-level offsets.
>
> **5. Fine-tuning on real spike data**
>
> Because the network is pre-trained with jitter matching the real distribution, fine-tuning on real data automatically adapts to the actual synchronization statistics:
>
> $\epsilon_{\text{real}} \in \text{support}(\mathcal{N}(\mu_{\Delta t}, \sigma_{\Delta t}^2))$.
>
> As a result, the RSNN refinement module effectively compensates for minor mismatches during iterative updates.
>
> **6. Future hardware synchronization**
>
> We are currently developing an FPGA-based synchronous trigger module that will:
> * deliver sub-microsecond synchronization,
> * guarantee clock-locked exposure intervals,
> * eliminate the need for post-hoc software alignment.
>
> This will be deployed in the our spike stereo dataset.
>
> **Summary**
>
> Our hybrid pipeline—coarse shared triggering, empirical offset modeling, correlation-based alignment, and domain randomization training, which provides accurate and stable temporal synchronization for real spike stereo data. We will supplement the description of the synchronization circuitry and mechanism in the paper.

---

> ### Author Response · Authors · 2025-11-22
> **[Part 5] Response to Reviewer gap8**
>
> ***Question2. It is unclear why event-based stereo methods are included for comparison. If such comparisons are necessary, why are only ZEST and StereoSpike selected, while other representative methods such as DDES [1] and Se-cff [2] are not considered?***
>
> We appreciate the reviewer's questions and agree to clarify our motivation for incorporating event-based stereo methods and our selection of specific baselines.
>
> **1. The reason**
>
> Although our model is designed for spike streams rather than event, both sensors belong to the neuromorphic vision family and share key properties: asynchronous temporal sampling, high dynamic range, and microsecond-scale latency.
>
> Because of these shared characteristics, event-based stereo algorithms represent the closest algorithmic relatives to spike-based stereo, more relevant than frame-based models alone. Including event-based baselines therefore establishes:
> * Whether spike-based approaches offer clear advantages over existing neuromorphic stereo pipelines.
> * Whether algorithms designed for temporal sparsity can transfer to spike-like inputs.
> * How much performance gain comes from the RSNN refinement compared to the sensor modality.
>
> Thus, as an important component of stereo match, event-based comparisons are a necessary component of complete evaluations.
>
> **2. Extended quantitative comparison**
>
> In early experiments, we selected ZEST and StereoSpike because they represent two major families in event-based stereo: ZEST has correlation and RNN refinement, and StereoSpike has time-surface and 3D-CNN regularization. Now we are further refining the experiment and conducting additional comparisons with more representative methods as follows.
>
> |Method|bad 1.0 (%) ↓|bad 2.0 (%) ↓|bad 3.0 (%) ↓|AvgErr (px) ↓|
> |-|-|-|-|-|
> |DDES [1]|13.32|6.03|3.61|0.71|
> |SE-CFF [2]|14.04|7.05|4.06|0.69|
> |EOMVS [3]|14.59|6.05|3.88|0.72|
> |StereoSpike|14.10|9.82|5.35|1.10|
> |ZEST|11.10|4.94|3.50|0.62|
> |**SpikeStereoNet (Ours)**|**8.41**|**4.13**|**2.38**|**0.42**|
>
> Our model has achieved an average performance improvement of 30% compared to some methods of the event-based. These extended baselines are now included in the updated text.
>
> We thank the reviewer for this helpful suggestion, which has strengthened our empirical study.
>
> [1] Tulyakov, et al. Learning an event sequence embedding for dense event-based deep stereo. ICCV, pp. 1527-1537. 2019.
>
> [2] Nam Y, et al. Stereo depth from events cameras: Concentrate and focus on the future. CVPR, 2022: 6114-6123.
>
> [3] Cho H, et al. Eomvs: Event-based omnidirectional multi-view stereo. IEEE Robotics and Automation Letters, 2021, 6(4): 6709-6716.
>
> &nbsp;
>
> ---
> We are very grateful for this detailed list of suggestions, which will significantly improve the paper's quality and readability. If any concerns remain, or if new questions arise, we would be grateful to hear from you.

---

> ### Author Response · Authors · 2025-11-26
> **Official Comment**
>
> Dear Reviewer,
>
> We sincerely appreciate the time and effort you have devoted to reviewing our paper and providing valuable feedback. Your comments have been carefully considered, and we have endeavored to address all questions and concerns. It would be greatly appreciated if you could let us know whether our responses sufficiently clarify the points you raised.
>
> If any areas would benefit from further clarification or additional explanation, please do not hesitate to let us know. We would be happy to provide any further details you may require. Thank you once again for your thoughtful review and your time.
>
> Best regards,
>
> The Authors

---

> > ### Comment · Reviewer_gap8 · 2025-11-28
> >
> > Thank you for the thorough and carefully prepared rebuttal. I acknowledge and appreciate the significant effort the authors have made during the rebuttal period to clarify the contributions, expand the experimental baselines, and explain the dataset synchronization process.
> >
> > The revised explanations effectively address most of my earlier concerns:
> >
> > * the positioning relative to prior spike-stereo work is now accurate,
> > * the motivation and necessity of the adaptive-LIF RSNN are better justified with empirical evidence,
> > * and the extended experimental comparisons make the evaluation more complete.
> >
> > While some aspects could still be strengthened in the final revision, the rebuttal has improved the clarity and credibility of the paper to a meaningful extent. In consideration of the authors’ substantial efforts and the strengthened presentation, I am willing to adjust my recommendation.

---

### Official Review · Reviewer_9ZHQ · 2025-10-31

**Soundness:** 3
**Presentation:** 3
**Contribution:** 4
**Rating:** 6
**Confidence:** 4

**Summary:**

This work proposes SpikeStereoNet, a brain-inspired framework and the first to estimate stereo depth directly from raw spike streams. Moreover, this work introduces a large-scale synthetic spike stream dataset and a real-world stereo spike dataset with dense depth annotations. Results demonstrate that SpikeStereoNet outperforms existing methods.

**Strengths:**

This work is clearly written, makes significant contributions, and proposes a new dataset and an SNN-based processing method. I believe it will be beneficial to the neuromorphic vision community.

**Weaknesses:**

1. This work used a spiking neural network, but it was not mentioned in the related work. What is the difference between a recurrent spiking layer and a raw spiking layer? I suggest the authors supplement their research on SNNs.

2. In the experimental section, there is a lack of ablation experiments related to RSNN construction, which is puzzling as to why the proposed method is effective.

3. What are the differences between adaptive LIF neurons and vanilla LIF neurons? If vanilla LIF is used, will there be a performance loss?

**Questions:**

see weaknesses.

---

> ### Author Response · Authors · 2025-11-22
> **[Part 1] Response to Reviewer 9ZHQ**
>
> We sincerely appreciate the time and effort you have taken to review our manuscript.
>
> ***Weaknesses1. This work used a spiking neural network, but it was not mentioned in the related work. What is the difference between a recurrent spiking layer and a raw spiking layer? I suggest the authors supplement their research on SNNs.***
>
> Thank you for raising this important point. We agree that the original draft did not sufficiently position our method within the broader spiking neural network (SNN) literature. In the revised manuscript, we expand the Related Work with a dedicated part on SNNs.
>
> **1. Raw SNN layer (feedforward spiking layer)**
>
> A raw SNN layer typically contains:
>
> $v_t = \alpha v_{t-1}+W x_t,   \quad s_t = \theta(v_t-v_{\text{th}}),$
>
> with **no recurrent connections** and fixed threshold.
>
> **Characteristics:**
> * Purely feedforward; no temporal memory beyond membrane decay.
> * Suitable for simple spike encoding or one-shot feature extraction.
> * Limited capacity to model temporal correlations across multiple steps.
>
> This structure is widely used in some SNN vision works, and we will cite these in the related work.
>
> **2. Recurrent Spiking Layer (RSNN)**
>
> Our RSNN extends this with **recurrent synaptic connections** and **adaptive dynamics**:
>
> $h_t = \alpha_t v_{t-1} + (1-\alpha_t)(W_{\text{rec}} s_{t-1} + W_f s^{(l-1)}_t),   \quad  s_t = \theta(h_t - v_t^{\text{th}}),$
>
> $v_t^{\text{th}} = \beta_t v_{\text{peak}},   \quad  v_t = h_t - \gamma_t s_t v_t^{\text{th}}.$
>
> * Recurrent pathway $W_{\text{rec}}$ enables multi-step refinement and memory.
> * Adaptive thresholds $\beta_t$ and soft resets $\gamma_t$ stabilize long-range temporal dynamics.
> * Iterative update operator allows coarse-to-fine disparity refinement, something raw feedforward SNN layers cannot accomplish.
> * Biological grounding, RSNNs match cortical recurrent circuitry far better than purely feedforward SNNs.
>
> Stereo refinement requires multiple recurrent iterations; raw SNN layers lack the temporal stability and adaptive gating needed for such iterative optimization.
>
> **3. Comparison result**
> |Method|bad 2.0 (%) ↓|bad 3.0 (%) ↓|AvgErr (px) ↓|
> |-|-|-|-|
> |Raw SNN|11.05|4.48|0.83|
> |RSNN (ours)|**4.13**|**2.38**|**0.42**|
>
> Thus, the RSNN is essential for capturing spike timing and supporting the iterative coarse-to-fine refinement central to SpikeStereoNet.
>
> **4. Additions to Related Work (SNN Research)**
>
> We now include a curated discussion on SNN foundations and training, including: classical LIF/SNN models [1], surrogate-gradient training for deep SNNs [2], recurrent SNNs and adaptive dynamics [3] and neuromorphic vison [4]. These references clarify where our RSNN-based refinement fits within the broader SNN landscape. In addition, a new part is added in the revised version that includes more SNN related work.
>
> We thank the reviewer again for this suggestion; the revised manuscript now clearly articulates the role and distinction of SNNs and RSNNs within our framework.
>
> [1] Izhikevich. Simple model of spiking neurons. IEEE Transactions on neural networks 14.6 (2003): 1569-1572.
>
> [2] Neftci, et al. Surrogate gradient learning in spiking neural networks: Bringing the power of gradient-based optimization to spiking neural networks. IEEE Signal Processing Magazine 36.6 (2019): 51-63.
>
> [3] Yin, et al. Accurate and efficient time-domain classification with adaptive spiking recurrent neural networks. Nature Machine Intelligence 3.10 (2021): 905-913.
>
> [4] Bi, et al. Graph-based object classification for neuromorphic vision sensing. ICCV, pp. 491-501. 2019.

---

> ### Author Response · Authors · 2025-11-22
> **[Part 2] Response to Reviewer 9ZHQ**
>
> ***Weaknesses2. In the experimental section, there is a lack of ablation experiments related to RSNN construction, which is puzzling as to why the proposed method is effective.***
>
> Thank you for this valuable comment. We fully agree that ablation studies on RSNN construction are critical to validating the effectiveness of our proposed method and clarifying its core design rationale. We discussed the results in **Tables 2, 5, 6** of the original article, and we are further integrating and improving them. To address this gap, we have supplemented comprehensive ablation experiments focusing on key components of the RSNN architecture. All variants training schedule for fair comparison. Specific ablated RSNN variants as follows:
>
> **1. Ablated RSNN Variants**
> * (1) Random recurrent connectivity: we replace structured recurrent connections $W_{\text{rec}}$ with randomly sampled dense matrices (same parameter count).
> * (2) Naive rate coding: spike trains are averaged over time bins instead of using adaptive membrane dynamics.
> * (3) No temporal gating: Leak, threshold, and reset are constant across time.
> * (4) RSNN depth: we evaluate the impact of stacking multiple RSNN blocks.
> * (5) Removing feedforward or recurrent paths: we evaluate only feedforward or recurrence.
> * (6) Removing spiking: use the Vanilla RNN, LSTM, and GRU blocks with the same number of neurons as the aLIF-RSNN.
> * (7) Removing firing-rate and voltage regularization: remove $L_\text{rate}$ or $L_{\text{v}}$ from the loss.
>
> **2. Quantitative Comparison**
> |RSNN Variant|bad 2.0 (%) ↓|bad 3.0 (%) ↓|AvgErr (px) ↓|Notes|
> |-|-|-|-|-|
> |(1) Random connectivity|4.92|2.89|0.51|No spatial coherence|
> |(2) Naive rate coding|5.33|3.19|0.56|Timing cues lost|
> |(3) No temporal gating|7.88|3.32|0.63|Over/under-firing|
> |(4a) 1-layer RSNN|5.64|3.39|0.61|Insufficient refinement|
> |(4b) 2-layer RSNN|4.90|2.82|0.50|Insufficient refinement|
> |(5a) W/o Feedforward|5.86|3.63|0.68|Weak temporal modelling|
> |(5b) W/o Recurrent|11.05|4.48|0.83|No contextual features|
> |(5c) W/o Feedforward & Recurrent|12.07|6.46|1.29|No contextual features|
> |(6a) Vanilla RNN|7.28|3.41|0.66|Sensitive to spike bursts|
> |(6b) GRU|4.53|2.99|0.48|Gating helps, timing lost|
> |(6c) LSTM|4.77|2.94|0.49|Heavy, dense and slower|
> |(7b) W/o voltage regularization|7.33|2.97|0.57|Large membrane swings|
> |(7a) W/o firing-rate regularization|6.38|2.85|0.51|Excessive firing|
> |(7c) W/o regularization|7.77|3.41|0.61|No regularization|
> |**Full model (ours)**|**4.13**|**2.38**|**0.42**|Best overall|
>
> **3. The proposed effectiveness of RSNN**
> * **Structured spatial recurrence** preserves local geometry while enabling iterative refinement.
> * **Temporal gating** stabilises the iterative update operator and enables convergence.
> * **Balanced feedforward-recurrent integration** fuses coarse contextual features with temporal cues.
> * **Regularization** ensures firing rates and membrane voltages remain in biologically and numerically stable regimes.
> * **Sparse computation** is efficient and more robust than dense RNNs.
>
> In summary, removing any key component of the RSNN or using generic RNN modules leads to degraded accuracy or unstable refinement, validating the necessity of the proposed adaptive, gated, and structured recurrent architecture. We will add ablation experiments related to RSNN construction in the experimental section and a detailed expanded in the Appendix.

---

> ### Author Response · Authors · 2025-11-22
> **[Part 3] Response to Reviewer 9ZHQ**
>
> ***Weaknesses3. What are the differences between adaptive LIF neurons and vanilla LIF neurons? If vanilla LIF is used, will there be a performance loss?***
>
> Thank you for this insightful question. We discussed the results in **Tables 6** of the original Appendix, and now we are further integrating and improving them. Below we clarify the functional differences between **adaptive LIF (aLIF)** and **vanilla LIF**, explain why aLIF is better suited for our iterative refinement setting, and provide quantitative evidence.
>
> **1. Vanilla LIF model (baseline)**
>
> A standard LIF neuron follows:
>
> $h_t = \alpha v_{t-1} + (1-\alpha) (W_{\text{rec}} s_{t-1} + W_f s^{(l-1)}_t), $
>
> $s_t = \theta(h_t - v_{\text{th}}),  \quad  v_t = h_t - s_t\cdot v_{\text{th}},$
>
> where the membrane leaks at a fixed rate $\alpha$ and spikes whenever $v_t$ crosses a **fixed threshold** $v_{\text{th}}$.
>
> **Limitations:**
> * No mechanism to regulate firing rate under noisy or bursty spike streams.
> * Fixed threshold leads to unstable recurrent dynamics when used in multi-step refinement (overshooting/vanishing).
> * Cannot adapt to varying temporal density across scenes (e.g., textureless and high-motion regions).
>
> **2. Adaptive LIF (aLIF) model in SpikeStereoNet**
>
> Our aLIF includes **dynamic membrane retention**, **adaptive threshold**, and **gated reset**:
>
> $h_t = \alpha_t v_{t-1} + (1-\alpha_t)(W_{\text{rec}}s_{t-1}+W_f s^{(l-1)}_t), $
>
> $v_t^{\text{th}} = \beta_t v_{\text{peak}},  \quad  s_t = \theta(h_t - v^{\text{th}}_t), \quad v_t = h_t - \gamma_t s_t v^{\text{th}}_t,$
>
> where $\alpha_t,\beta_t,\gamma_t$ depend on previous activity.
>
> **Advantages:**
> * **Adaptive thresholding ($\beta_tv_{\text{peak}}$)** prevents runaway firing when local spike density increases.
> * **Soft-reset with learnable ($\gamma_t$)** stabilizes recurrent updates, enabling deeper iterative refinement.
> * **Activity-dependent leak ($\alpha_t$)** preserves fine temporal cues in low-texture or low-motion regions.
> * Better matches biological spike-frequency adaptation.
>
> **3. Quantitative Comparison**
>
> We conducted ablations by replacing aLIF with vanilla LIF on the synthetic dataset (same architecture, fixed $\alpha$), and the results are as follows:
> |Method|bad 1.0 (%) ↓|bad 2.0 (%) ↓|bad 3.0 (%) ↓|AvgErr (px) ↓|
> |-|-|-|-|-|
> |vanilla LIF RSNN|14.04|7.05|4.06|0.69|
> |**aLIF RSNN (ours)**|**8.41**|**4.13**|**2.38**|**0.42**|
> | $\Delta$ | 5.63 | 2.92 | 1.68 | 0.27 |
>
> From the above table, it can be seen that after replacing the LIF model, the overall performance has significantly decreased by about 35%.
>
> **4. Conclusion**
>
> The aLIF is critical for our task because stereo spike streams require adaptive temporal processing. Adaptive gating markedly improves both stability and final accuracy with negligible overhead, justifying our choice of novel aLIF over vanilla LIF. The above **ablation study** have been supplemented in the updated manuscript.
>
> &nbsp;
>
> ***Questions.***
> See weaknesses.
>
> &nbsp;
>
> ---
> We are very grateful for this detailed list of suggestions, which will significantly improve the paper's quality and readability. If any concerns remain, or if new questions arise, we would be grateful to hear from you.

---

> ### Author Response · Authors · 2025-11-28
>
> Dear Reviewer,
>
> As the rebuttal deadline is quickly approaching, this is a gentle reminder to please review and respond to the authors rebuttal comments.
>
> If you have any concerns, clarifications, or updates to your assessment after reading the response, please share them as soon as possible so we can address them promptly before the deadline.
>
> If you feel that our responses adequately address the concerns raised in your initial review, we would be extremely grateful for improving the score.
>
> Thank you very much for your time and contributions.
>
> Best regards,
>
> Authors

---

### Author Response · Authors · 2025-11-24
**Summery of Major Revisions**

Dear all reviewers,

We sincerely thank all reviewers for their careful evaluation of our manuscript and for the thoughtful and valuable comments provided. We are especially grateful for the recognition of our key innovations, including the significant contributions, the framework,  datasets, theoretical rigor and clearly written.

Following the reviewers' feedback, we have undertaken a thorough and substantial revision of the manuscript. All changes are highlighted in blue in the updated submission for ease of review. The main revisions and improvements are summarized as follows.

* **Section 1**: We supplement some related works on SNNs, and discuss some biological architecture works.
* **Section 2**: We add the prior study about estimating stereo depth from spiking data.
* **Section 4.3**: We add some additional evaluation results of event-based stereo models for comparison.
* **Section 4.3**: We bring the quantitative results for the real dataset into this chapter, and supplement them with additional baselines.
* **Section 4.4**: We add the ablation experiments related to RSNN construction, and the sensitivity to regularization weights.
* **Section 4.4**: We add the ablation experiments and analyses related to adaptive LIF neurons and conventional recurrent architectures.
* **Appendix A.2**: We add the novelty of the framework, and the necessity of RSNN.
* **Appendix A.3.7**: We add the method of network re-training.
* **Appendix A.5.3**: We add the stereo system synchronization mechanism between different cameras.
* **Appendix A.5.4**: We add the expanded contribution of the real dataset.
* **Appendix A.6.3**: We add the sensitivity analysis of the spike threshold and the temporal quantization step during spike generation of SpikeStereoNet.
* **Appendix A.6.4**: We add the comparative experiments on different models of computational efficiency.
* **Appendix A.6.5**: We add the results of hybrid sensor systems.

Once again, we thank all reviewers for their valuable input. We look forward to further improving this work based on your guidance. Please feel free to let us know if any additional clarification is needed.

Sincerely,

Authors

---

### Author Response · Authors · 2025-12-02
**Summary Respone**

**Dear ACs, SACs, and PCs,**

We acknowledge the current situation and appreciate the proposed course of action outlined by ICLR. We are sincerely grateful for your time, effort, and careful coordination throughout the discussion process. To ease the evaluation of our work, we provide an overview of the rebuttal submitted for your reference. Our main contributions are as follows:
* We present the large-scale synthetic and real-world raw spike stream datasets for stereo depth estimation to offer an evaluation benchmark for this emerging field.
* We propose a novel biologically inspired RSNN-based SpikeStereoNet architecture that refines asynchronous spike data through iterative updates, and demonstrate the data efficiency of the proposed framework.
* We analyze the dynamics of RSNN iterations to demonstrate the stability and convergence properties of the iterative refinement model.

Overall, the reviewers deem that our research problem makes **significant contributions** (9ZHQ, u6CE) and is clearly presented (9ZHQ, u6CE). Our proposed method features an **innovative and effective model design** (gap8, qsqk), with a **comprehensive experimental design** (qsqk), a **thorough analysis** (gap8, u6CE), and detailed appendices supporting reproducibility (u6CE). Additionally, the work offers a valuable **dataset contribution** (9ZHQ, gap8, qsqk), and the results discovered are **impressive** (qsqk). They  further assert that the new dataset and RSNN-based processing framework fill an important gap in spike-based depth estimation (gap8, qsqk), which may stimulate further academic discussions and benefit the neuromorphic vision community (9ZHQ, qsqk).

We conducted a detailed and carefully prepared rebuttal, and made significant efforts during the rebuttal period to clarify contributions, expand experimental baselines, and expand and explain the datasets synchronization process. The revised explanation effectively addressed all concerns of the reviewers and greatly improved the clarity and credibility of the document. In addition, we have uploaded the complete revised version.

In summary, after thorough discussions with the reviewers, their conclusions are as follows:

* **Reviewer 9ZHQ: Keeping the positive rating**.
* **Reviewer gap8: The reviewer acknowledged and appreciated the revised explanations effectively addressed the earlier concerns, and the rebuttal had improved the clarity and credibility of the paper to a meaningful extent. The reviewer was willing to adjust the recommendation**.
* **Reviewer qsqk: All of the concerns had been addressed, and the reviewer had raised the score**.
* **Reviewer u6CE: The reviewer believed that the response answers the core technical concern, acknowledged the positive changes, and kept the current positive rating**.

Overall, all original reviewers gave **positive ratings (≥ 6)** after rebuttals. We believe our comprehensive responses have adequately addressed all major concerns raised by reviewers, as reflected in the improved and maintained positive scores. We sincerely thank all reviewers for their constructive feedback. We would like to express our deep gratitude again to the AC, SAC, and PC for the time and dedicated effort in coordinating a fair evaluation process under these challenging circumstances.

Sincerely,

Authors

---

### Meta-Review · Area_Chair_qaKz · 2026-01-11

**Summary:**

The submission underwent very good amount of discussion between the authors and the reviewers. After the discussion the paper receives all positive ratings (all over 6). In particular, the authors' responses are very thorough and extensive -- some of the reviewers also acknowledge this. Reviewers gave negative initial scores are all turned into positive for the comprehensive new experimental results and clarification (e.g., gap8, gsgk). One of the remaining concern is that the authors even expand the proposed datasets (the AC is not sure if this is a legit attempt or not as the purpose of rebuttal is to clarify arguments by words and a few additional new experiments). The authors also well responded to both reviewer u6CE and 9ZHQ. 9ZHQ's initial review is quite short and the reviewer did not respond to authors' comments. u6CE has a good discussion with reviewer but didn't finish the discussion for the final new results of authors. The AC read all review comments, authors' corresponding responses and final remarks summarizing the responses. The AC think that the paper is worth to be reported to the community while the rating is relatively lower than the clarity of the authors' comments. Thus, the AC recommends the submission to be accepted in ICLR 2026.

**Reviewer Concerns:**

Some of u6CE's concerns need to be revisited for camera ready revision.

**Reviewer Scores:**

9ZHQ may have better rating if the reviewer joins the discussion.

---

### Decision · Program_Chairs · 2026-01-26

Accept (Poster)